

# HOPE: An Arbitrary-Order Non-Oscillatory Finite-Volume Shallow
# Water Dynamical Core with Automatic Differentiation
Lilong Zhou[1,2,3] Wei Xue[1]
*1.   Department of Computer Science and Technology, Tsinghua University, Beijing, 100084, China*
*2.   Department of Model Technology, CMA Earth System Modeling and Prediction Centre (CEMC), Beijing, 100081,*
*China*
*3.   State Key Laboratory of Severe Weather, Beijing, 100081, China*
**Corresponding author:** Wei Xue(xuewei@tsinghua.edu.cn)
**Key words:** Automatic differentiation; Arbitrary-order accuracy; Non-Oscillation; Finite-volume methods; Shallow water
Equations; Dynamical core.
# Abstract
This study presents the High Order Prediction Environment (HOPE), an automatically differentiable, non-oscillatory
finite-volume dynamical core for shallow water equations on the cubed-sphere grid. HOPE integrates four key features: (1)
arbitrary high-order accuracy through genuine two-dimensional reconstruction schemes; (2) essential non-oscillation via
adaptive polynomial order reduction in discontinuous regions; (3) exact mass conservation inherited from finite-volume
discretization; (4) automatically differentiable and (5) GPU-native scalability through PyTorch-based implementation.
Another innovation is the intensive panel boundary treatment, which eliminates numerical instability during using high order
reconstruction scheme, meanwhile, simplifies the interpolation process to a matrix-vector multiplication without losing
accuracy. Numerical experiments demonstrates the capabilities of HOPE: The 11th-order scheme reduces errors to near
double-precision round-off levels in steady-state geostrophic flow tests on coarse $1° \times 1°$ grids. Maintenance of Rossby-
Haurwitz waves over 100 simulation days without crashing. A cylindrical dam-break test case confirms the genuinely two-
dimensional WENO scheme exhibits significantly better isotropy compared to dimension-by-dimension approaches. Two
implementations are developed: a Fortran version for convergence analysis and a PyTorch version leveraging automatic
differentiation and GPU acceleration. The PyTorch implementation maps reconstruction and quadrature operation to 2D
convolution and Einstein summation respectively, achieving about $2\times$ speedup on single NVIDIA RTX3090 GPU versus
Dual Intel E5-2699v4 CPUs execution. This design enables seamless coupling with neural network parameterizations,
positioning HOPE as a foundational tool for next-generation differentiable atmosphere models.



## 1. Introduction

Recent years have witnessed a surge in research integrating numerical weather prediction (NWP) with artificial
intelligence (AI) techniques. A prominent advancement in this domain is the hybrid modeling paradigm, which synergizes the
complementary strengths of both approaches. This framework implements numerical dynamical cores within AI software
platforms such as TensorFlow or PyTorch, thereby enabling seamless integration of AI models into the numerical solution
process for atmospheric dynamical partial differential equations (PDEs). Unlike the fully surrogated methods, such as Pangu-
Weather (Bi et al., 2022), FengWu (Chen et al., 2023), GraphCast (Lam et al., 2023), NowcastNet (Zhang et al., 2023). Hybrid
model integrates traditional PDE-based dynamical cores with neural network (NN)-based physical parameterizations. The
auto-differentiable nature of the dynamical core enables training losses to propagate through the entire model during
backpropagation, allowing the NN-based parameterization module to access more comprehensive residual information.
NeuralGCM (Kochkov et al., 2023) exemplifies this hybrid approach by combining a spectral numerical dynamical core with
NN-based physical parameterizations. The governing equation-based dynamical core imposes rigorous physical constraints
within the framework, effectively mitigating the blurriness characteristic of purely data-driven models. Furthermore,
NeuralGCM demonstrates superior power spectra performance compared to conventional data-driven meteorological models.
While the implementation of a spectral dynamical core in NeuralGCM theoretically enables infinite-order accuracy,
the inherent shortcomings of the spectral model still persist. Specifically, it fails to preserve mass conservation, and the global
nature of spectral expansion also restricts the scalability of this method.
To address these shortcomings, we present the High Order Prediction Environment (HOPE) dynamical core with
following contributions:
1)    A new-generation shallow-water model architecture integrating:

(i)    Arbitrary high-order accuracy (up to 13th-order verified) via tensor product polynomial (TPP).

(ii)    Local stencil-based operations enabling massively parallel scalability.

(iii)    Inherent mass conservation from finite-volume discretization.

(iv)    Adaptive polynomial order reduction for essential non-oscillation.

2)    A novel intensive ghost cell interpolation scheme achieving:

(i)    Arbitrary odd-order convergence through central stencil interpolation.

(ii)    Single sparse matrix-vector operation replacing iterative procedures (Appendix Eq.(A.12)).

(iii)    Overcome numerical instability beyond 7th-order accuracy.

3)    PyTorch-based high performance differentiable implementation featuring:

(i)    GPU acceleration through convolution/einsum operator in PyTorch, 2× speedup on single RTX3090 GPU vs.



59  Dual Intel Xeon 2699v4 CPUs.

60  (ii) Automatic Jacobian matrix generation via native auto-differentiation.

61  (iii) Seamless integration with NN modules for hybrid modeling.

62  In the following part of the introduction, we introduce the relevant work on constructing the HOPE model, and from this,

63 we elaborate on the challenges and motivations for establishing the algorithm of the dynamical core. High-order accuracy is

64 an extremely appealing trait for the design of a dynamical core, particularly in high-resolution atmospheric simulations. A

65 dynamical core model with high-order accuracy produces significantly less simulation error in smooth regions compared to a

66 low-order model. Furthermore, even when the resolution is equivalent or coarser, a high-order model is capable of resolving

67 finer details than a low-order one.

68  A high-order finite volume model was developed on cubed sphere, named MCORE (Ullrich et al., 2010; Ullrich and

69 Jablonowski, 2012). The authors assert that MCORE's convergence accuracy can theoretically be of arbitrary order. However,

70 in the practical numerical tests, we found that the accuracy does not surpass the 7th order. This limitation arises when using a

71 one-sided ghost interpolation scheme, which leads to numerical oscillations originating from the corner zones of the panels

72 when the stencil size is 9×9 or larger.

73  In this article, we devise the reconstruction based on tensor product polynomial (TPP). When the stencil width is $k$, our

74 method achieves $k^{th}$ order accuracy, surpassing MCORE by one order of accuracy with the same stencil width. In addition,

75 we have developed a new class of ghost interpolation schemes that abandon the use of one-sided stencils and instead adopt

76 central stencils. This new approach enables the scheme to overcome the non-physical oscillations arising from interpolation

77 at panel boundaries. Our method allows for arbitrary accuracy, and we have verified this by testing up to the 11[th] order.

78  From the properties of the Taylor series, we note that its effectiveness in approximating a function depends on two key

79 conditions: (1) the existence of higher-order derivatives, and (2) the convergence of the series. When the field exhibits poor

80 continuity—where higher-order derivatives either do not exist or lead to increasing residuals with series order—employing

81 higher-order approximations can introduce significant errors. Therefore, for reconstruction schemes based on polynomial

82 functions, high-order accuracy should only be adopted when the field is sufficiently smooth. Conversely, for discontinuous or

83 poorly continuous fields, reducing the reconstruction order is necessary to maintain numerical stability and effectiveness.

84  The Weighted Essentially Non-Oscillatory (WENO) scheme is an adaptive limiter widely employed in computational

85 fluid dynamics (CFD) to address this challenge. Originally developed for one-dimensional problems (Liu et al., 1994), WENO

86 was later extended to two dimensions by Shi et al. (2002) using two distinct approaches: a genuinely two-dimensional

87 (WENO2D) scheme and a dimension-by-dimension reconstruction. In this work, we implement WENO2D scheme to enforce

88 the non-oscillatory property. This approach effectively suppresses non-physical oscillations near sharp discontinuities while

89 preserving high accuracy in smooth regions.





The remainder of this paper is organized as follows: Section 2 details the governing equations on the cubed-sphere grid.
Section 3 presents the numerical methods, including reconstruction schemes, panel boundary treatment method, and temporal
marching scheme. Section 4, describes the GPU-optimized implementation leveraging PyTorch's automatic differentiation.
Section  5 validates model performance through standard test cases, followed by conclusions and future directions in Section

6.

# 2. Governing Equation on Cubed Sphere

The cubed-sphere grid partitions the spherical domain into six panels, each with a structured and rectangular
computational space. This configuration facilitates high-order spatial reconstruction and efficient massive-thread parallelism
(see Figure 1). Early work on solving the primitive equations on the cubed-sphere grid dates back to Sadourny (1972). In
recent decades, the cubed-sphere grid has been widely adopted in high-order-accuracy atmospheric models. For instance,
Chen and Xiao (2008) developed a shallow water model using the multi-moment constrained finite volume method on the
cubed sphere, achieving $3^{rd}{\sim}4^{th}$ order accuracy. Ullrich et al. (2010) designed a high-order finite volume dynamical core based
on this grid, Nair et al. (2005a, 2005b) implemented a discontinuous Galerkin model on the cubed sphere.
In this study, we also employ the cubed-sphere grid. Although the mesh is non-orthogonal, the computational space can
still be treated as a rectangular grid by adopting a generalized curvilinear coordinate system. In this section, we present the
shallow water equations in generalized curvilinear coordinates and discuss specialized treatments for topography.

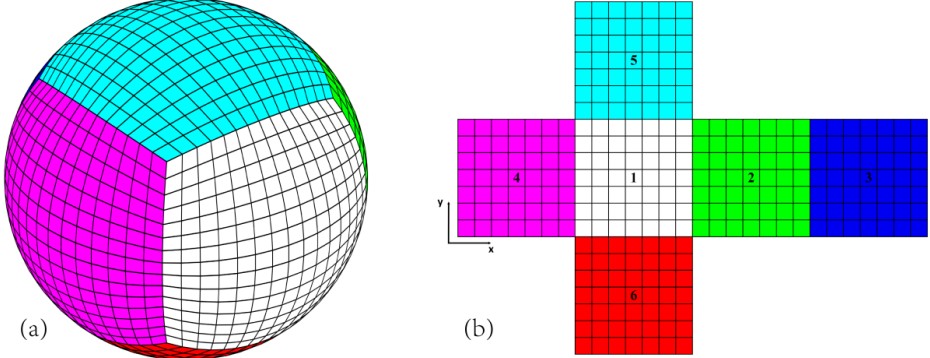


**Figure 1** Cubed sphere grid. (a) Physical space; (b) Computational space. Six panels are identified by indices from 1 to 6.
Shallow water equation set on gnomonic equiangular cubed sphere grid is written as

$$\begin{cases} \dfrac{\partial \sqrt{G}\phi}{\partial t} + \dfrac{\partial \sqrt{G}\phi u}{\partial x} + \dfrac{\partial \sqrt{G}\phi v}{\partial y} = 0 \\[3mm] \dfrac{\partial \sqrt{G}\phi u}{\partial t} + \dfrac{\partial \sqrt{G}\left(\phi uu + \frac{1}{2}G^{11}\phi^2\right)}{\partial x} + \dfrac{\partial \sqrt{G}\left(\phi uv + \frac{1}{2}G^{12}\phi^2\right)}{\partial y} = \psi_M^1 + \psi_C^1 + \psi_B^1 \\[3mm] \dfrac{\partial \sqrt{G}\phi v}{\partial t} + \dfrac{\partial \sqrt{G}\left(\phi uv + \frac{1}{2}G^{21}\phi^2\right)}{\partial x} + \dfrac{\partial \sqrt{G}\left(\phi vv + \frac{1}{2}G^{22}\phi^2\right)}{\partial y} = \psi_M^2 + \psi_C^2 + \psi_B^2 \end{cases} \tag{1}$$





The gnomonic equiangular coordinates are represented by $(x, y, n_p)$, where $(x, y) \in \left[-\frac{\pi}{4}, \frac{\pi}{4}\right]$ are local equiangular coordinate
of each panel and $n_p$ is panel number. $\phi = gh$ is geopotential height, $h$ is fluid thickness, $u, v$ is contravariant wind in $x, y$
direction, $g$ is gravity acceleration. $\psi_M, \psi_C, \psi_B$ are the metric term, Coriolis term and bottom topography influence term

$$\psi_M = \begin{pmatrix} \psi_M^1 \\ \psi_M^2 \end{pmatrix} = \frac{2\sqrt{G}}{\delta^2} \begin{pmatrix} -XY^2\phi uu + Y(1+Y^2)\phi uv \\ X(1+X^2)\phi uv - X^2Y\phi vv \end{pmatrix} \tag{2}$$


$$\psi_C = -\sqrt{G}\sqrt{G}f\boldsymbol{k} \times \phi\boldsymbol{u} = \sqrt{G}f \begin{pmatrix} -G^{12} & G^{11} \\ -G^{22} & G^{12} \end{pmatrix} \begin{pmatrix} \sqrt{G}\phi u \\ \sqrt{G}\phi v \end{pmatrix} \tag{3}$$


$$\psi_B = -\sqrt{G}\phi G^{ij}\frac{\partial \phi_s}{\partial x^j} = -\sqrt{G}\phi \begin{pmatrix} G^{11}\dfrac{\partial \phi_s}{\partial x} + G^{12}\dfrac{\partial \phi_s}{\partial y} \\ G^{21}\dfrac{\partial \phi_s}{\partial x} + G^{22}\dfrac{\partial \phi_s}{\partial y} \end{pmatrix} \tag{4}$$

where $X = \tan x$, $Y = \tan y$, $\delta = \sqrt{1 + X^2 + Y^2}$, $f = 2\Omega \sin\theta$ is Coriolis parameter, $\phi_s = gh_s$ is surface geopotential
height, and $h_s$ is surface height.

$$\sin\theta = \begin{cases} Y/\delta, & n_p \in \{1,2,3,4\} \\ 1/\delta, & n_p = 5 \\ -1/\delta, & n_p = 6 \end{cases} \tag{5}$$

The contravariant metric on cubed-sphere is

$$G^{ij} = \frac{\delta^2}{r^2(1+X^2)(1+X^2)} \begin{pmatrix} 1+Y^2 & XY \\ XY & 1+X^2 \end{pmatrix} \tag{6}$$

The covariant metric

$$G_{ij} = \frac{r^2(1+X^2)(1+Y^2)}{\delta^4} \begin{pmatrix} 1+X^2 & -XY \\ -XY & 1+Y^2 \end{pmatrix} \tag{7}$$

and the metric determinant is given by

$$\sqrt{G} = \sqrt{\det(G_{ij})} = \frac{r^2(1+X^2)(1+Y^2)}{\delta^3} \tag{8}$$

$r$ is radius of earth.
The contravariant wind vector $\boldsymbol{V} = (u, v)$ can be convert to wind vector on spherical LAT/LON coordinate $\boldsymbol{V}_s = (u_s, v_s)$
by the following formula

$$\begin{pmatrix} u_s \\ v_s \end{pmatrix} = J \begin{pmatrix} u \\ v \end{pmatrix} \tag{9}$$

where $J$ is a $2 \times 2$ conversion matrix, the expressions are different in each panel





$$J = r \begin{pmatrix} \cos\theta \dfrac{\partial\lambda}{\partial x} & \cos\theta \dfrac{\partial\lambda}{\partial y} \\ \dfrac{\partial\theta}{\partial x} & \dfrac{\partial\theta}{\partial y} \end{pmatrix} = \begin{cases} r\begin{pmatrix} \cos\theta & 0 \\ -\sin\theta\cos\theta\tan\lambda_p & \cos\lambda_p\cos^2\theta + \dfrac{\sin^2\theta}{\cos\lambda_p} \end{pmatrix}, & panel\ 1\sim4 \\[2em] r\begin{pmatrix} \cos\lambda\sin\theta\ \Gamma_1 & \sin\lambda\sin\theta\ \Gamma_2 \\ -\sin\lambda\sin^2\theta\ \Gamma_1 & \cos\lambda\sin^2\theta\ \Gamma_2 \end{pmatrix}, & panel\ 5 \\[2em] r\begin{pmatrix} -\cos\lambda\sin\theta\ \Gamma_1 & \sin\lambda\sin\theta\ \Gamma_2 \\ \sin\lambda\sin^2\theta\ \Gamma_1 & \cos\lambda\sin^2\theta\ \Gamma_2 \end{pmatrix}, & panel\ 6 \end{cases} \tag{10}$$

$$\lambda_p = \lambda - \frac{\pi}{2}(i_{panel} - 1), \qquad \Gamma_1 = 1 + \frac{\sin^2\lambda}{\tan^2\theta}, \qquad \Gamma_2 = 1 + \frac{\cos^2\lambda}{\tan^2\theta} \tag{11}$$

where $\lambda, \theta$ are longitude and latitude, and $i_{panel}$ is the panel index as shown in Figure 1(b). The relation between $J$ and $G_{ij}$ is

$$G_{ij} = J^T J \tag{12}$$

In our numerical experiments, topography causes non-physical oscillation while we using equation set Eq.(1) and
reconstructing $\sqrt{G}\phi$, as mentioned by Chen and Xiao (2008), so called "C-property" needs to be preserved. Inspired by Ii and
Xiao (2010), we reconstruct $\sqrt{G}\phi_t$ instead of $\sqrt{G}\phi$, where $\phi_t = \phi + \phi_s$ is total geopotential height, and the reconstruction
method is introduced in the next section. The momentum equations need to be modified as follow

$$\begin{cases} \dfrac{\partial\sqrt{G}\phi}{\partial t} + \dfrac{\partial\sqrt{G}\phi u}{\partial x} + \dfrac{\partial\sqrt{G}\phi v}{\partial y} = 0 \\[1.2em] \dfrac{\partial\sqrt{G}\phi u}{\partial t} + \dfrac{\partial\sqrt{G}\left(\phi uu + \frac{1}{2}G^{11}\phi_t^2\right)}{\partial x} + \dfrac{\partial\sqrt{G}\left(\phi uv + \frac{1}{2}G^{12}\phi_t^2\right)}{\partial y} = \psi_M^1 + \psi_C^1 + \psi_B^1 \\[1.2em] \dfrac{\partial\sqrt{G}\phi v}{\partial t} + \dfrac{\partial\sqrt{G}\left(\phi uv + \frac{1}{2}G^{21}\phi_t^2\right)}{\partial x} + \dfrac{\partial\sqrt{G}\left(\phi vv + \frac{1}{2}G^{22}\phi_t^2\right)}{\partial y} = \psi_M^2 + \psi_C^2 + \psi_B^2 \end{cases} \tag{13}$$

and the bottom topography influence term is now expressed as

$$\psi_B = \sqrt{G}\phi_s G^{ij}\frac{\partial\phi_t}{\partial x^j} = \sqrt{G}\phi_s \begin{pmatrix} G^{11}\dfrac{\partial\phi_t}{\partial x} + G^{12}\dfrac{\partial\phi_t}{\partial y} \\ G^{21}\dfrac{\partial\phi_t}{\partial x} + G^{22}\dfrac{\partial\phi_t}{\partial y} \end{pmatrix} \tag{14}$$

The reconstruction variables are $\left(\sqrt{G}\phi_t, \sqrt{G}\phi u, \sqrt{G}\phi v\right)$.
We write the governing equation set to vector form

$$\frac{\partial \boldsymbol{q}}{\partial t} + \frac{\partial \boldsymbol{F}(\boldsymbol{q})}{\partial x} + \frac{\partial \boldsymbol{G}(\boldsymbol{q})}{\partial y} = \boldsymbol{S}(\boldsymbol{q}) \tag{15}$$

$$\boldsymbol{q} = \begin{bmatrix} \sqrt{G}\phi \\ \sqrt{G}\phi u \\ \sqrt{G}\phi v \end{bmatrix}, \boldsymbol{F} = \begin{bmatrix} \sqrt{G}\phi u \\ \sqrt{G}\left(\phi uu + \frac{1}{2}G^{11}\phi_t^2\right) \\ \sqrt{G}\left(\phi uv + \frac{1}{2}G^{21}\phi_t^2\right) \end{bmatrix}, \boldsymbol{G} = \begin{bmatrix} \sqrt{G}\phi v \\ \sqrt{G}\left(\phi uv + \frac{1}{2}G^{12}\phi_t^2\right) \\ \sqrt{G}\left(\phi vv + \frac{1}{2}G^{22}\phi_t^2\right) \end{bmatrix}, \boldsymbol{S} = \begin{bmatrix} 0 \\ \psi_M^1 + \psi_C^1 + \psi_B^1 \\ \psi_M^2 + \psi_C^2 + \psi_B^2 \end{bmatrix} \tag{16}$$

## 3. Numerical Discretization


The finite volume method computes the temporal tendency of cell-averaged quantities by evaluating the net flux across
cell interfaces. The interfacial flux is obtained through Gaussian quadrature, where the field values at quadrature points are





reconstructed spatially and then processed by a Riemann solver to determine the flux magnitude.
In this section, we present two distinct spatial reconstruction approaches: (1) a two-dimensional tensor product
polynomial (TPP) method, and (2) a two-dimensional weighted essentially non-oscillatory (WENO2D) scheme based on
tensor product polynomials. Each reconstruction yields two potential values at every Gaussian quadrature point (GQP). These
values are then resolved into a single flux value using the Low Mach number Approximate Riemann Solver (LMARS) (Chen
et al., 2013). Finally, the total flux across each cell edge is computed by applying linear Gaussian quadrature integration along
the interface.

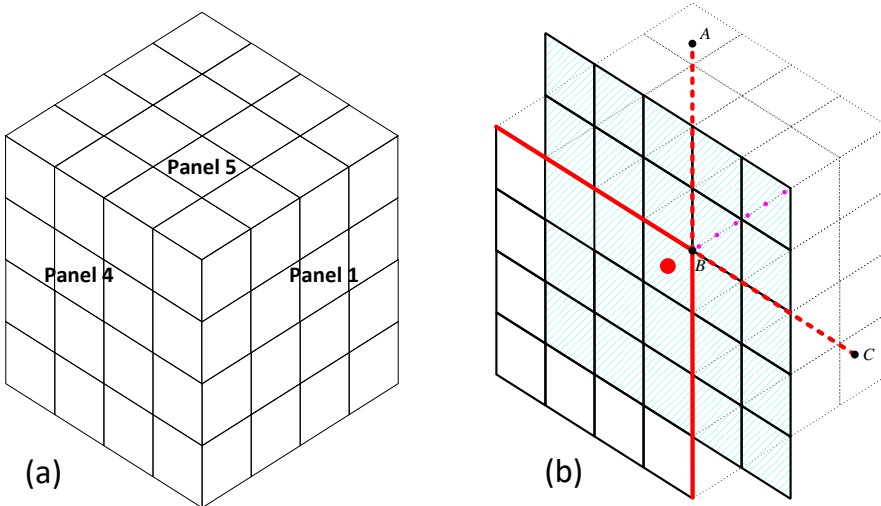


**Figure 2** (a) Adjacent area of panels 1,4 and 5. (b) $5 \times 5$ reconstruction stencil nearby panel corner is represented by shade. The
cell contains red dot is the target cell on panel 4, red solid lines are boundary of panel 4, red dash lines are extension line of panel
4 boundary line. $A$ and $C$ are points on dash line, $B$ is the upper right corner point of panel 4.
According to the finite volume scheme, average Eq.(15) on cell $i, j$, we have

$$\frac{\partial \overline{\boldsymbol{q}}_{i,j}}{\partial t} + \frac{\overline{\boldsymbol{F}}_{i+\frac{1}{2},j} - \overline{\boldsymbol{F}}_{i-\frac{1}{2},j}}{\Delta x} + \frac{\overline{\boldsymbol{G}}_{i+\frac{1}{2},j} - \overline{\boldsymbol{G}}_{i-\frac{1}{2},j}}{\Delta y} = \overline{\boldsymbol{S}}_{i,j} \tag{17}$$

$$\frac{\partial \overline{\boldsymbol{q}}_{i,j}}{\partial t} = \frac{1}{\Delta x \Delta y} \frac{\partial}{\partial t} \iint_{\Omega_{i,j}} \boldsymbol{q} \, dx \, dy \,, \qquad \overline{\boldsymbol{S}}_{i,j} = \frac{1}{\Delta x \Delta y} \iint_{\Omega_{i,j}} \boldsymbol{S} \, dx \, dy \tag{18}$$

$$\overline{\boldsymbol{F}}_{i-\frac{1}{2},j} = \frac{1}{\Delta y} \int_{e_{i-\frac{1}{2}}} \boldsymbol{F} \, dy, \qquad \overline{\boldsymbol{F}}_{i+\frac{1}{2},j} = \frac{1}{\Delta y} \int_{e_{i+\frac{1}{2}}} \boldsymbol{F} \, dy \tag{19}$$

$$\overline{\boldsymbol{G}}_{i,j-\frac{1}{2}} = \frac{1}{\Delta x} \int_{e_{j-\frac{1}{2}}} \boldsymbol{G} \, dx, \qquad \overline{\boldsymbol{G}}_{i,j+\frac{1}{2}} = \frac{1}{\Delta x} \int_{e_{j+\frac{1}{2}}} \boldsymbol{G} \, dx \tag{20}$$

where $\Omega_{i,j}$ represents the region overlapped by cell $(i,j)$, $e_{i-\frac{1}{2}}, e_{i+\frac{1}{2}}, e_{j-\frac{1}{2}}, e_{j+\frac{1}{2}}$ are left, right, bottom, top edges of cell $(i,j)$.



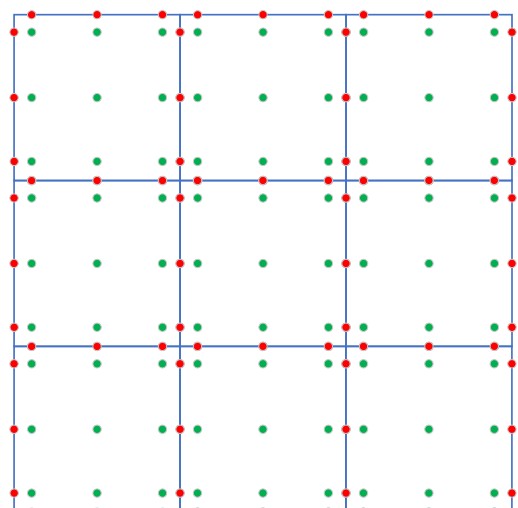


**Figure 3** Function points on cell. Red points are edge quadrature points (EQP) or called flux points, green points are inner cell

quadrature points (CQP).

The physical interpretation of equation Eq.(17) is that the average tendency of prognostic field $q$ within cell $(i, j)$ is

governed by the average net flux and average source. In this study, we calculate these averages using Gaussian quadrature,
the function points within each cell are illustrated in Figure 3, the EQPs are share by adjacent cells, and CQPs are exclusive
for each cell.

Average on edge by 1D scheme:

$$\overline{F}_{i+\frac{1}{2},j} \approx \sum_{r=1}^{m_e} w_r F_r = w F_r \tag{21}$$

where $w = \left(w_1, w_2, \dots, w_{m_e}\right)$ is the 1D Gaussian quadrature coefficient matrix, $m_e$ is the number of quadrature points on each
edge.

Average in cell by 2D scheme:

$$\overline{S}_{i,j} \approx \sum_{r=1}^{m_c} W_r S_r = W S_r \tag{22}$$

where $W = \left(W_1, W_2, \dots, W_{m_c}\right)$ is the 2D Gaussian quadrature coefficient matrix, $m_e$ is the number of quadrature points on
each cell.

## 3.1 Tensor Product Polynomial (TPP) Reconstruction

The computational space of cubed sphere is rectangular and structured, we take reconstruction on square stencil. A two-

dimensional $d$-th degree polynomial has number of terms $n = \frac{(d+1)(d+2)}{2}$, it is not able to be fully filled by a $k$-th order square



stencil ($k \times k$ cells), as shown in Figure 4 (a). The stencil location on cubed sphere grid is shown in Figure 2(b)
(a)                    (b)                    (c)
**Figure 4** Polynomial terms on stencils. (a): 2nd degree polynomial stencil; (b): 3rd order TPP stencil; (c) 5th order TPP stencil
We make use of the TPP to approximate the horizontal reconstruction. A TPP is expressed as

$$p(x,y) = \sum_{i=1}^{m}\sum_{j=1}^{n} a_k x^{i-1} y^{j-1} = \sum_{k=1}^{N} a_k c_k(x,y) \tag{23}$$

where $n$ is width of stencil (also called $n$-th stencil). $a_k$ is the coefficient of each term, the term index $k = i + n(j-1)$, and
$c_k(x,y) = x^\alpha y^\beta, \alpha = k - int\left(\frac{k-1}{n}\right)n - 1, \beta = int\left(\frac{k-1}{n}\right)$, $int$ is equivalent to Fortran's intrinsic function $int()$ that
truncates to integer values. $m = n^2$ is the cell number in stencil and also the term number of the TPP, the 3rd and 5th order
stencils are shown in Figure 4. We define column vectors $c(x,y) = \{c_k(x,y)|k = 1,2,3,\dots,N\}$ and $a = \{a_k|k =$
$1,2,3,\dots,N\}$, the point value on $(x,y)$ can be written as

$$p(x,y) = c(x,y) \cdot a \tag{24}$$

The volume integration average (VIA) of prognostic field $q$ on cell $\Omega_i$ is represented by

$$\bar{q}_i = \frac{1}{\Delta x_i \Delta y_i} \iint_{\Omega_i} p(x,y) dx dy \tag{25}$$

$\Delta x_i, \Delta y_i$ are length of edges $x, y$ of cell $\Omega_i$ in computational space. In our setting, all of the cells in the computational
space are set to unit square, therefore $\Delta x_i = 1, \Delta y_i = 1$, and Eq.(25) becomes

$$\bar{q}_i = \iint_{\Omega_i} p(x,y) dx dy = \iint_{\Omega_i} c \cdot a \, dx dy = \psi_i \cdot a \tag{26}$$

where $\psi_i = \iint_{\Omega_i} c \, dx dy = \begin{pmatrix} \iint_{\Omega_i} c_1 dx dy \\ \iint_{\Omega_i} c_2 dx dy \\ \vdots \\ \iint_{\Omega_i} c_N dx dy \end{pmatrix}$, combining $N$ cells, we have following linear system

$$A a = \bar{q} \tag{27}$$

$$A = \begin{pmatrix} \psi_1^T \\ \psi_2^T \\ \vdots \\ \psi_N^T \end{pmatrix}, \bar{q} = \begin{pmatrix} \bar{q}_1 \\ \bar{q}_2 \\ \vdots \\ \bar{q}_N \end{pmatrix} \tag{28}$$

and polynomial coefficient $a$ can be obtain by solving Eq.(27).

$$a = A^{-1}\bar{q} \tag{29}$$

The reconstruction values on $M$ points can be obtained by following formula



$$p = Ca = CA^{-1}\overline{q} = R\overline{q} \tag{30}$$

where $p = \begin{pmatrix} p(x_1, y_1) \\ p(x_2, y_2) \\ \vdots \\ p(x_M, y_M) \end{pmatrix}, C = \begin{pmatrix} c_1^T \\ c_2^T \\ \vdots \\ c_M^T \end{pmatrix}, c_j^T = c^T(x_j, y_j), j = 1,2, \dots, M$ , superscript $T$ stands for transpose matrix,
$(x_j, y_j)$ represents the function points on target cell. The reconstruction matrix

$$R = CA^{-1} \tag{31}$$

The reconstruction matrix $R$ needs to be computed only once during model initialization and stored in memory. In
practical implementation, the TPP reconstruction procedure can be directly formulated as a two-dimensional convolutional
operation using $R$ as the convolution kernel.

## 3.2 Genuine Two-Dimensional WENO

Weighted Essentially Non-Oscillatory (WENO) represents an adaptive algorithm that dynamically preserves high-order
approximation accuracy in smooth flow regions while automatically degenerating to robust low-order reconstruction near
discontinuities for effective shock capturing. Shi et al. (2002) mentioned two different approaches for constructing a fifth-
order finite volume WENO scheme: the "Genuine 2D" method and the "Dimension by Dimension" method.
For HOPE, within the Genuine 2D framework:
a) Third-order reconstruction utilizes a $3\times3$ cell stencil that decomposes into four $2\times2$ sub-stencils
b) Fifth-order accuracy employs a $5\times5$ master stencil partitioned into nine distinct $3\times3$ sub-stencils
(Complete schematic representations of these decomposition strategies are provided in Figure 5 and Figure 6)
The scheme's theoretical accuracy fundamentally depends on the proper determination of optimal linear weights for the
multidimensional stencil combination. These weights, when correctly derived, enable the weighted superposition of sub-
stencils to recover full high-order accuracy in smooth solution regions. While (Shi et al., 2002) indicated the theoretical
possibility of computing these weights through Lagrange interpolation basis analysis, they omitted specific implementation
details. In this section, we present the methods for constructing genuine 2D WENO (WENO 2D) schemes using least squares
method.





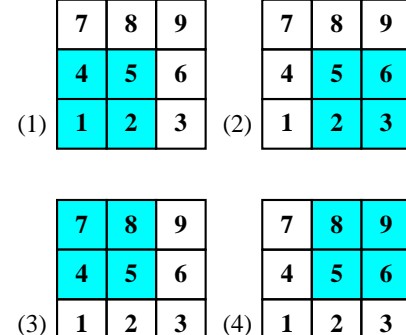


**Figure 5** Stencils of 3$^{rd}$ order WENO 2D. The high order stencil contains cells No.1~9, blue ones represent the cells in sub-
stencils (1) ~ (4).

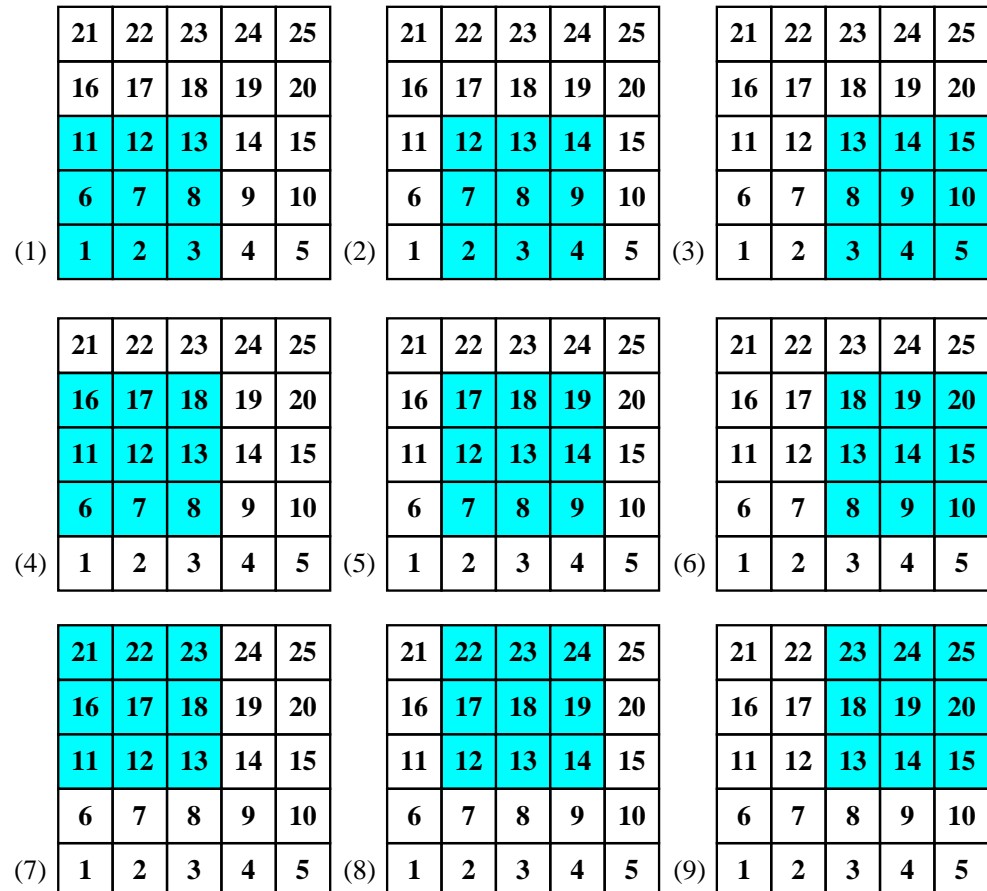


**Figure 6** Stencils of 5$^{th}$ order WENO 2D. The high order stencil contains cells No.1~25, blue ones represent the cells in sub-
stencils (1) ~ (9).

We construct WENO 2D based on TPP and square stencil. As mentioned in previous section, a $n$-th order stencil contains





$m = n^2$ cells, and the stencil width is $h = n$. Decomposing the high-order stencil into $s = \left(\frac{n+1}{2}\right)^2$ sub-stencils, there are $s_c =$
$s$ cells in each sub-stencil, and the sub-stencil width is $l = \frac{n+1}{2}$. For the reconstruction point $(x, y)$, suppose $p_H(x, y)$ is the
reconstruction value of high-order stencil, the reconstruction values of sub-stencils are stored in vector $\boldsymbol{p} =$
$\left(p_1(x, y), p_2(x, y), \cdots, p_s(x, y)\right)^T$. The intention of constructing the optimal linear weights is to determine the unique weights
$\boldsymbol{\gamma} = (\gamma_1, \gamma_2, \cdots, \gamma_s)$, such that

$$p_H = R_H \overline{\boldsymbol{q}} = \boldsymbol{\gamma p} \tag{32}$$

where the elements of vector $\overline{\boldsymbol{q}} = (q_1, q_2, \cdots, q_m)^T$ represent VIA of each cell in high-order stencil. $R_H = \left(r_{H_j}\right), j =$
$1, 2, \dots, m$ is the reconstruction matrix of high-order stencil.
The computation of $\gamma$ requires the integration of both high-order and low-order reconstruction matrices into a unified
linear system. For each sub-stencil $i$ we define the reconstruction matrix $R_i = (r_{ik}), k = 1, 2, \dots, s_c$ (computed via Eq.(31)).
and $R_{L_i} = \left(r_{L_{ij}}\right), j = 1, 2, \dots, m$ is the extension matrix of $R_i$. The matrix relationship is expressed as
$$(R_i)_{1 \times s_c}(E)_{s_c \times m} = \left(R_{L_i}\right)_{1 \times m}$$
where the subscripts denote matrix dimensions. The correspondence matrix $E = \left(e_{ij}\right), i = 1, 2, \dots, s_c; j = 1, 2, \dots, m$ encodes
the cell relationships between stencils: when the $i$-th cell in low-order stencil is the same as the $j$-th cell in high order stencil,
$e_{ij} = 1$, otherwise, $e_{ij} = 0$.
Substitute Eq.(30) into Eq.(32), yield

$$R_H \overline{\boldsymbol{q}} = \sum_{i=1}^{s} R_{L_i} \gamma_i \overline{\boldsymbol{q}} \tag{33}$$

We set $R_L = \left(R_{L_1}, R_{L_2}, \dots, R_{L_s}\right)^T$, Eq.(33) becomes

$$R_L \gamma = R_H \tag{34}$$

The unknown optimal weight matrix $\gamma$ can be determined by following least square procedure

$$\gamma = (R_L^T R_L)^{-1} R_L^T R_H \tag{35}$$

However, the elements of $\gamma$ could be negative, which would cause unstable. A split technique mentioned by (Shi et al.,
2002) was adopted to solve this problem. The optimal weights can be split into two parts:

$$\gamma^+ = \frac{\theta|\gamma| + \gamma}{2}, \qquad \gamma^- = \gamma^+ - \gamma \tag{36}$$

where the constant $\theta = 3$. The reconstruction value on point $(x, y)$ is expressed by:

$$q(x, y) = \sum_{i=1}^{s} (\omega_i^+ - \omega_i^-) p_i(x, y) \tag{37}$$

The nonlinear weights $\omega_i$ is large when stencil $i$ is smooth on target cell and if stencil $i$ is discontinuous, $\omega_i$ should be a small
value. Several nonlinear weighting schemes have been developed to meet these criteria, including WENO-JS (Jiang and Shu,



1996), WENO-Z (Borges et al., 2008), WENO-Z+ (Acker et al., 2016), WENO-Z+M (Luo and Wu, 2021), among others.
In this work, we employ the WENO-Z formulation as our baseline scheme. While most existing WENO schemes were
originally developed for one-dimensional problems, we propose a two-dimensional extension of WENO-Z through
modification of $\tau$, a crucial coefficient that governs the scheme's higher-order accuracy properties.
For stencil $i$ the nonlinear weight is given as

$$\omega_i^\pm = \frac{\alpha_i^\pm}{\sum_{i=1}^s \alpha_i^\pm} \tag{38}$$

$$\alpha_i^\pm = \gamma_i^\pm \left(1 + \frac{\tau}{\beta_i + \varepsilon}\right) \tag{39}$$

$$\tau = \frac{2}{(s-1)s} \sum_{\eta=1}^{s-1} \sum_{\psi=\eta+1}^{s} |\beta_\psi - \beta_\eta| \tag{40}$$

The smooth indicators $\beta_i$ measure how smooth the reconstruction functions are in the target cell; we use a similar scheme
as described in Zhu and Shu (2019):

$$\beta_j = \sum_{\zeta=1}^{m} \iint_\Omega \frac{\partial^\zeta}{\partial x^{\zeta_1} \partial y^{\zeta_2}} p_j(x,y) dx dy \tag{41}$$

where $\zeta_1 + \zeta_2 = \zeta$ and $\zeta > 0$, $\zeta_1, \zeta_2 \in [0, n]$.

## 3.3 Treatment of the Panel Boundaries

The cubed sphere grid comprises eight panel boundaries, and the flux across the interface between any two panels must
be computed at the quadrature points situated on the edges of the boundary cells, as depicted in Figure 7 (a). However, a
challenge arises because the coordinates across these panel boundaries are discontinuous. Given that the TPP reconstruction
necessitates a square stencil, the values of the cells outside the domain (referred to as ghost cells) must be computed through
interpolation within the adjacent panel, as illustrated in Figure 7 (b). Ullrich et al. (2010) proposed a one-side interpolation
scheme, but in our test, we found that using one-sided interpolation around panel boundaries leads to instability when the
accuracy exceeds the 7th order.





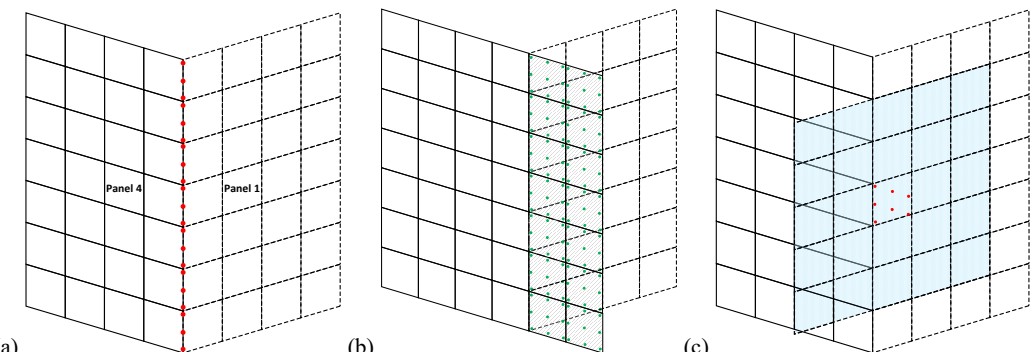

(a)                          (b)                          (c)

**Figure 7** Points and cells close to panel boundary. (a) Flux points on the interface between Panel 1 and Panel 4, the flux across
each panel at these points are determined by Riemann solver, which merges the reconstruction outcomes from both panels into a
single flux value; (b) Ghost cells (shaded cells) out of Panel 4 boundary, with green points representing the GQPs in these cells;
(c) Cells requirement for 5th order ghost cell interpolation stencil, red points represent the GQPs located in the ghost cell on Panel
4, the blue shaded region represents the TPP reconstruction stencil (on Panel 1) to interpolate these red GQPs.

### 3.3.1 Ghost Cell Interpolation

To achieve arbitrary high-order accuracy, we propose a ghost cell interpolation scheme that incorporates information
from both sides of the panel boundary. Since the ghost cell values are inherently unknown prior to interpolation, our approach
involves an initial estimation through an iterative process. Specifically, the method iteratively performs ghost cell interpolation
until the increments of the cell values converge to within a specified tolerance.

Through mathematical analysis (detailed in the Appendix), we demonstrate that this iterative process can be expressed
as a linear mapping, thereby eliminating the need for actual iterations. However, direct computation of the mapping matrix
requires inversion of a large matrix, which poses significant computational and memory challenges. To address this, we
implement the iterative interpolation process using PyTorch and leverage its automatic differentiation capability to efficiently
obtain the interpolation matrix.

The complete methodology, as derived in the Appendix, proceeds as follows:

1.  Initialization: All ghost cell values are initialized to zero (denoted as $g^{(0)} = 0$, where the superscript indicates the
    iteration number).

2.  Interpolation: The Gaussian quadrature points (GQPs) in the ghost cells are interpolated using the Taylor polynomial
    preserving (TPP) stencil. For instance, considering two adjacent panels (Figure 7(a)), any out-domain cell in Panel 4
    (shaded cell in Figure 7(b)) contains GQPs that physically reside in Panel 1. These GQPs are interpolated using the TPP
    stencil shown in Figure 7(c), which incorporates relevant ghost cells from Panel 1.

3.  Update and convergence check: After interpolating all GQPs, the ghost cell values are updated via Gaussian quadrature
    (Eq. (22)), yielding $\boldsymbol{g}^{(1)}$. The L2-norm residual $r^{(k)} = \left\| \boldsymbol{g}^{(k+1)} - \boldsymbol{g}^{(k)} \right\|_2$ is then computed. Steps 2-3 repeat until $r^{(k)} <$





$\epsilon$, where $\epsilon = 1.e^{-14}$ for double precision and $\epsilon = 1.e^{-5}$ for single precision. In practice, convergence typically occurs
within 10 iterations, so we fix the iteration count at 10 for consistency.
This process establishes a linear mapping $G: \boldsymbol{q} \to \boldsymbol{g}$ from known cell values to ghost cell values. As proven in Eq.(A.12)
(Appendix), the mapping's linearity implies that $G = \frac{\partial \boldsymbol{g}}{\partial \boldsymbol{q}}$ forms a matrix, which we efficiently compute using PyTorch's
autograd functionality. This approach avoids explicit matrix inversion while maintaining numerical precision.

### 3.3.2   Fields Conversion Between Panels

Due to the differing coordinate systems across panels, field variables must be appropriately transformed when
transferring information between adjacent panels. To illustrate this process, we consider the interface between Panel 1 and
Panel 4, as depicted in Figure 2(a) and Figure 7(a). Although flux points are shared between the two panels, their coordinate
representations are discontinuous across the interface.
To ensure consistency, two key transformations are required:
1.   Metric reset for mass variables: The mass-related prognostic quantities must be recomputed in the target panel's
coordinate system to maintain metric consistency.
2.   Wind vector transformation: Velocity components (or other vector quantities) must be converted from the source
panel's local coordinate frame to that of the target panel.
This coordinate conversion ensures proper continuity and physical consistency when interpolating or exchanging data
across panel boundaries.
Suppose $\boldsymbol{q}_1 = \left[ (\sqrt{G}\phi)_1, (\sqrt{G}\phi u)_1, (\sqrt{G}\phi v)_1 \right]^T$ and $\boldsymbol{q}_4 = \left[ (\sqrt{G}\phi)_4, (\sqrt{G}\phi u)_4, (\sqrt{G}\phi v)_4 \right]^T$ represent the fields on
panel 1 and 4. The mass field conversion from panel 4 to panel 1 is expressed by

$$\left( \sqrt{G}\phi \right)_4^1 = \frac{\sqrt{G}_4}{\sqrt{G}_1} \left( \sqrt{G}\phi \right)_1 \tag{42}$$

the subscript represents the target panel and the superscript stands for source panel.
The transformation of momentum vectors between panels is performed in two sequential steps to maintain proper tensor
consistency. The contravariant momentum components in Panel 1 are first projected onto the global spherical coordinate
system using the transformation matrix $J$, as defined in Eq.(10). The resulting spherical momentum components are then
transformed into the contravariant representation specific to Panel 4, ensuring compatibility with the target panel's local
coordinate system.

$$\begin{bmatrix} (\sqrt{G}\phi u_s)_1 \\ (\sqrt{G}\phi v_s)_1 \end{bmatrix} = J_1 \begin{bmatrix} (\sqrt{G}\phi u)_1 \\ (\sqrt{G}\phi v)_1 \end{bmatrix} \tag{43}$$



$$\begin{bmatrix} (\sqrt{G}\phi u)_4 \\ (\sqrt{G}\phi v)_4 \end{bmatrix} = J_4^{-1} \frac{\sqrt{G}_4}{\sqrt{G}_1} \begin{bmatrix} (\sqrt{G}\phi u_s)_1 \\ (\sqrt{G}\phi v_s)_1 \end{bmatrix} \tag{44}$$

where $J_1$ is the $J$ matrix on panel 1, $J_4^{-1}$ is the inverse matrix of $J$ on panel 4, $(u_s, v_s)$ are zonal wind and meridional wind,
$(u, v)$ are contravariant wind components. Since the vector conversion is linear process, Eq.(43) and Eq.(44) can be merged
into following equation

$$\begin{bmatrix} (\sqrt{G}\phi u)_4 \\ (\sqrt{G}\phi v)_4 \end{bmatrix} = C \begin{bmatrix} (\sqrt{G}\phi u)_1 \\ (\sqrt{G}\phi v)_1 \end{bmatrix} \tag{45}$$

where matrix $C = \frac{\sqrt{G}_4}{\sqrt{G}_1} J_4^{-1} J_1$.
The mass and vector are also need to be converted on GQPs in the same manner.

## 3.4 Riemann Solver

Following spatial reconstruction, discontinuous solutions arise on either side of each flux point location, since the
majority of atmospheric flow speeds correspond to small Mach numbers, we adopt the Low Mach number Approximate
Riemann Solver (Chen et al., 2013) as Riemann solver to determine the flux at the edge quadrature points (EQPs).
Spatial reconstruction gives the left and right state vector,

$$\boldsymbol{q}_L = \begin{bmatrix} (\sqrt{G}\phi)_L \\ (\sqrt{G}\phi u)_L \\ (\sqrt{G}\phi v)_L \end{bmatrix}, \qquad \boldsymbol{q}_R = \begin{bmatrix} (\sqrt{G}\phi)_R \\ (\sqrt{G}\phi u)_R \\ (\sqrt{G}\phi v)_R \end{bmatrix} \tag{46}$$

First of all, we convert the contravariant wind $u$ to physical speed $u^\perp$ that is perpendicular to the cell edge.

$$u^\perp = \frac{u}{\sqrt{G^{ii}}}, \qquad i = \begin{cases} 1, & x\ direction \\ 2, & y\ direction \end{cases} \tag{47}$$

The wind speed $m^*$ and geopotential height $\phi$ are calculated by

$$m^* = \frac{1}{2}\left(u_L^\perp + u_R^\perp - \frac{\phi_R - \phi_L}{c}\right) \tag{48}$$

$$\phi = \frac{1}{2}[\phi_L + \phi_R - c(u_R^\perp - u_L^\perp)] \tag{49}$$

$$c = \frac{c_L + c_R}{2} \tag{50}$$

$$c_L = \sqrt{\phi_L}, c_R = \sqrt{\phi_R} \tag{51}$$

$c$ is the phase speed of external gravity wave, the subscript $L, R$ represent the left and right side of cell edge.
Once $m^*$ is determined, we convert it back to contravariant speed by

$$m = m^* \sqrt{G^{ii}} \tag{52}$$

The flux across the cell edge is then given by

$$\boldsymbol{F} = \frac{1}{2}[m(\boldsymbol{q}_L + \boldsymbol{q}_R) - sign(m)(\boldsymbol{q}_R - \boldsymbol{q}_L)] + \boldsymbol{P} \tag{53}$$





$$P = \begin{pmatrix} 0 \\ \frac{1}{2}\sqrt{G}G^{1i}\phi_t^2 \\ \frac{1}{2}\sqrt{G}G^{2i}\phi_t^2 \end{pmatrix}, \qquad i = \begin{cases} 1, & x\ direction \\ 2, & y\ direction \end{cases} \tag{54}$$

For calculation of $\boldsymbol{H}$ the method is similar.

## 3.5 Temporal Integration

We use the explicit Runge-Kutta (RK) as time marching scheme, Wicker and Skamarock (2002) described a 3rd order
RK with three stages, for the prognostic fields $\boldsymbol{q}$, the integration step from time slot $n$ to $n + 1$:

$$\boldsymbol{q}^* = \boldsymbol{q}^n + \frac{\Delta t}{3}\left(\frac{\partial \boldsymbol{q}^n}{\partial t}\right) \tag{55}$$

$$\boldsymbol{q}^{**} = \boldsymbol{q}^* + \frac{\Delta t}{2}\left(\frac{\partial \boldsymbol{q}^*}{\partial t}\right) \tag{56}$$

$$\boldsymbol{q}^{n+1} = \boldsymbol{q}^n + \Delta t\left(\frac{\partial \boldsymbol{q}^{**}}{\partial t}\right) \tag{57}$$

where $\Delta t$ is the time step, and temporal tendency terms $\frac{\partial \boldsymbol{q}}{\partial t}$ can be obtain by Eqs.(15), (16).

# 4. High Performance Implementation and Automatic Differentiation

The spatial operator and temporal integration of HOPE can be easily implemented using PyTorch. Specifically, the spatial
reconstruction given by Eq.(30) is analogous to a convolution operation, while the Gaussian quadrature can be efficiently
expressed as a matrix-vector multiplication. Both of these operations are highly optimized for execution on GPUs, ensuring
superior performance. Furthermore, as a versatile platform for AI development, PyTorch offers automatic differentiation
capabilities for all the aforementioned functions, streamlining the implementation and enabling efficient gradient
computation.
For the reconstruction implementation. Suppose the cubed sphere grid comprises $n_c$ cells in $x$-direction within each
panel, including ghost cells. The panel number is $n_p$, and the shallow water equation involves $n_v$ prognostic variables, we
write the cell state tensor $\boldsymbol{q}$ with the shape $(n_v n_p, 1, n_c, n_c)$. The TPP reconstruction weight tensor $\boldsymbol{R}$ has shape
$(n_{poc}, 1, s_w, s_w)$, where $n_{poc}$ is the number of points required to be interpolated within each cell (including EQP and CQP),
$s_w$ denotes the stencil width. The reconstruction can be executed by a simple command (pseudo-code):

$$\boldsymbol{q}_{rec} = torch.nn.Functional.conv2d(\boldsymbol{q}, \boldsymbol{R}) \tag{58}$$

where the shape of $\boldsymbol{q}_{rec}$ is $(n_v n_p, n_{poc}, n_c, n_c)$
For the Gaussian quadrature implementation. Suppose the edge state tensor $\boldsymbol{q}_e$ with the shape $(n_v, n_p, n_c, n_c, n_{poe})$,
where $n_{poe}$ is the number of quadrature points on each edge. The edge Gaussian quadrature weight tensor $\boldsymbol{g}_e$ has shape





$(n_{poe})$. The quadrature is expressed by:

$$\boldsymbol{q}_g = torch.matmul(\boldsymbol{q}_e, \boldsymbol{g}_e) \tag{59}$$

where the shape of $\boldsymbol{q}_g$ is $(n_v, n_p, n_c, n_c)$
After spatial reconstruction, the resulting data is utilized in the Riemann solver for EQPs and for source term computation
on CQPs. Subsequently, integration is performed on both EQPs and CQPs to calculate the net flux and the cell-averaged
source term tendency. However, there is a dimensionality mismatch between the reconstructed points, i.e. $n_{poc}$ is the first
dimension of $\boldsymbol{q}_{rec}$, while $n_{poe}$ is the last dimension of $\boldsymbol{q}_e$. To address this dimensionality issue, two methods are available.
The first method involves rearranging the $n_{poc}$ dimension to the last position using the "torch.tensor.permute" operation in
PyTorch, This allows Gaussian integrations to be naturally implemented through the "torch.matmul" operation. The second
method, which avoids the need for the "permute" operation, maintains the original dimension sequence. Instead, Gaussian
integrations are performed using the "torch.einsum" function. This function sums the product of the elements of the input
arrays along dimensions specified using a notation based on the Einstein summation convention.

$$\boldsymbol{q}_g = torch.\,einsum('vnpij, p \rightarrow vnij', \boldsymbol{q}_e, \boldsymbol{g}_e) \tag{60}$$

We have conducted performance tests comparing the two methods, and the results indicate that the second method is
approximately 5% faster than the first. Specifically, the first method took 649 seconds, while the second method took 616
seconds. The test was set as a one-day steady state geostrophic flow (with details provided in section 5.1) simulation at a
resolution of $0.1°$, using $3^{rd}$ order accuracy reconstruction stencil. The time step was 30 seconds, and the default data type
used was "torch.float32" (single precision).
The Riemann solver implementation on flux points is way easier, only needs to call "torch.sign" for Eq.(53), while all
other operations can be executed using basic arithmetic: addition, subtraction, multiplication, and division. During a Runge-
Kutta sub-step, there are no dependencies, and neither "for" loops nor "if" statements are required in the HOPE kernel code.
This algorithm fully leverages the capabilities of the GPU.

## 5. Numerical Experiments

The HOPE dynamical core is evaluated using the standard test cases (Test 2, 5, and 6) for the spherical shallow water
model as described in Williamson et al. (1992), along with the perturbed jet flow case proposed by Galewsky et al. (2004).
Additionally, a dam-break experiment is designed to demonstrate the HOPE model's capability in capturing shock waves.
In our experiments, the grid resolutions are denoted by the count of cells along one dimension on each panel of the cubed
sphere; for instance, C90 signifies that each panel is subdivided into a $90 \times 90$ grid, corresponding to a grid interval of $\Delta x =$
$\Delta y = 1°$.





## 5.1 Steady State Geostrophic Flow

Steady state geostrophic flow is the 2[nd] case in Williamson et al. (1992), it provided an analytical solution for spherical shallow water equations, it was wildly used in accuracy test for shallow water models. The analytical solution is a steady state, which means the initial filed is the exact solution. The initial field is expressed as

$$\phi = \phi_0 - \left(a\Omega u_0 + \frac{u_0^2}{2}\right)(-\cos\lambda\cos\theta\sin\alpha + \sin\theta\cos\alpha)^2 \tag{61}$$

$$u_s = u_0(\cos\theta\cos\alpha + \cos\lambda\sin\theta\sin\alpha) \tag{62}$$

$$v_s = -u_0\sin\lambda\sin\alpha \tag{63}$$

where $\lambda, \theta$ are longitude and latitude, $\phi$ is geopotential height, $u_s, v_s$ are zonal wind and meridional wind, earth radius is $a = 6371220\ m$, earth rotation angular velocity $\Omega = 7.292 \times 10^{-5}\ s^{-1}$, basic flow speed $u_0 = \frac{2\pi a}{12*86400}\ m/s$, basic geopotential height $\phi_0 = 29400\ m^2/s^2$, $\alpha = 0$ and gravity acceleration $g = 9.80616\ m/s^2$. The conversion between the spherical wind $(u_s, v_s)$ and contravariant wind is given by Eq.(9).

We use three kinds of norm errors to measure the simulation errors,

$$L_1 = \frac{I\big[\phi(i,j,p) - \phi_{ref}(i,j,p)\big]}{I\big[\phi_{ref}(i,j,p)\big]} \tag{64}$$

$$L_2 = \sqrt{\frac{I\left[\big(\phi(i,j,p) - \phi_{ref}(i,j,p)\big)^2\right]}{I\big[\phi_{ref}^2(x,y,p)\big]}} \tag{65}$$

$$L_\infty = \frac{\max\big|\phi(i,j,p) - \phi_{ref}(i,j,p)\big|}{\max\big|\phi_{ref}(i,j,p)\big|} \tag{66}$$

$$I(\phi) = \sum_{p=1}^{n_p}\sum_{j=1}^{n_y}\sum_{i=1}^{n_x}\big(\sqrt{G}\phi\big)_{ijp} \tag{67}$$

where $n_x, n_y$ represent the number cells in $x, y$ directions, and $n_p = 6$ is the number of panels on cubed sphere grid. The metric Jacobian $\sqrt{G}$ has the same definition as Eq.(8). For example, a C90 grid corresponds $n_x = n_y = 90$.

We simulated the steady state geostrophic flow over one period (12 days) to test the norm errors and corresponds convergence rate. Since the norm error becomes too small to express by double precision number, all of the experiments were based on the quadruple precision version of HOPE. Time steps were set to $\Delta t = 600, 400, 200, 100, 50\ s$ for C30, C45, C90, C180 and C360, respectively.

**Table 1** Norm errors and convergence rates of steady state geostrophic flow at day 12.

| TPP3 | C30 | C45 | C90 | C180 | C360 |
|---|---|---|---|---|---|
| $L_1$ error | 1.8853E-03 | 5.6474E-04 | 7.0960E-05 | 8.8777E-06 | 1.1099E-06 |
| $L_1$ rate | | 2.9731 | 2.9925 | 2.9988 | 2.9998 |
| $L_2$ error | 2.1484E-03 | 6.4171E-04 | 8.0500E-05 | 1.0069E-05 | 1.2588E-06 |
| $L_2$ rate | | 2.9802 | 2.9949 | 2.9991 | 2.9998 |
| $L_\infty$ error | 4.3242E-03 | 1.2932E-03 | 1.6201E-04 | 2.0275E-05 | 2.5350E-06 |





| | | | | | |
|---|---|---|---|---|---|
| $L_\infty$ rate | | 2.9770 | 2.9968 | 2.9983 | 2.9997 |
| **TPP5** | | | | | |
| $L_1$ error | 3.6122E-06 | 4.7493E-07 | 1.4827E-08 | 4.6322E-10 | 1.4474E-11 |
| $L_1$ rate | | 5.0039 | 5.0014 | 5.0004 | 5.0002 |
| $L_2$ error | 5.2427E-06 | 6.9169E-07 | 2.1627E-08 | 6.7584E-10 | 2.1119E-11 |
| $L_2$ rate | | 4.9954 | 4.9992 | 5.0000 | 5.0001 |
| $L_\infty$ error | 1.6810E-05 | 2.2451E-06 | 7.0534E-08 | 2.2070E-09 | 6.8985E-11 |
| $L_\infty$ rate | | 4.9652 | 4.9923 | 4.9982 | 4.9996 |
| **TPP7** | | | | | |
| $L_1$ error | 8.1697E-08 | 4.7967E-09 | 3.7678E-11 | 2.9547E-13 | 2.3125E-15 |
| $L_1$ rate | | 6.9922 | 6.9922 | 6.9946 | 6.9974 |
| $L_2$ error | 8.7991E-08 | 5.1644E-09 | 4.0507E-11 | 3.1728E-13 | 2.4823E-15 |
| $L_2$ rate | | 6.9931 | 6.9943 | 6.9963 | 6.9979 |
| $L_\infty$ error | 1.4741E-07 | 8.6376E-09 | 6.7814E-11 | 5.3387E-13 | 4.1901E-15 |
| $L_\infty$ rate | | 6.9971 | 6.9929 | 6.9889 | 6.9934 |
| **TPP9** | | | | | |
| $L_1$ error | 7.8909E-10 | 2.1780E-11 | 4.3925E-14 | 8.6359E-17 | |
| $L_1$ rate | | 8.8537 | 8.9538 | 8.9905 | |
| $L_2$ error | 9.5638E-10 | 2.6409E-11 | 5.3341E-14 | 1.0494E-16 | |
| $L_2$ rate | | 8.8526 | 8.9516 | 8.9896 | |
| $L_\infty$ error | 2.3946E-09 | 6.6773E-11 | 1.3547E-13 | 2.6644E-16 | |
| $L_\infty$ rate | | 8.8285 | 8.9452 | 8.9899 | |
| **TPP11** | | | | | |
| $L_1$ error | 1.1908E-10 | 1.3799E-12 | 6.7696E-16 | 3.3197E-19 | |
| $L_1$ rate | | 10.9943 | 10.9932 | 10.9938 | |
| $L_2$ error | 1.3084E-10 | 1.5186E-12 | 7.4489E-16 | 3.6500E-19 | |
| $L_2$ rate | | 10.9904 | 10.9934 | 10.9949 | |
| $L_\infty$ error | 2.4204E-10 | 2.8579E-12 | 1.4147E-15 | 6.9567E-19 | |
| $L_\infty$ rate | | 10.9479 | 10.9803 | 10.9898 | |


In Table 1, we present the geopotential height simulation errors and convergence accuracy of different order accuracy

schemes at various resolutions. It is evident that HOPE is capable of achieving the designed accuracies in all tests. When the
resolution exceeds C180, the errors obtained from the 7th, 9th, and 11th-order precision schemes have surpassed the limits
expressible by double-precision numbers. This demonstrates HOPE's excellent error convergence for simulating smooth flow
fields. It should be noted that high-order accuracy schemes do consume more computational resources. HOPE has proven the
feasibility of ultra-high-order accuracy finite volume methods on cubed sphere grids. However, in simulating the real
atmosphere, a balance between computational efficiency and error must be considered. We believe that $3^{rd}$ or $5^{th}$ order
accuracy schemes will be more practical for subsequent developments in baroclinic atmosphere model.

## 5.2 Zonal Flow over an Isolated Mountain

Zonal flow over an isolated mountain is the $5^{th}$ case mentioned in Williamson et al. (1992), this case was usually be

implemented to test the topography influence in shallow water models. The initial condition is defined by Eq.(61)~(63), but



$h_0 = 5960\,m$, $\phi_0 = h_0 g$, $u_0 = 20 m/s$. The mountain height is expressed as

$$h_s = h_{s0}\left(1 - \frac{r}{R}\right) \tag{68}$$

where $h_{s0} = 2000\,m$; $R = \frac{\pi}{9}$; $r = \sqrt{\min[R^2, (\lambda - \lambda_c)^2 + (\theta - \theta_c)^2]}$, $\lambda_c, \theta_c$ are the center longitude and latitude of the
mountain, respectively, we set $\lambda_c = \frac{3\pi}{2}$, $\theta_c = \frac{\pi}{6}$.

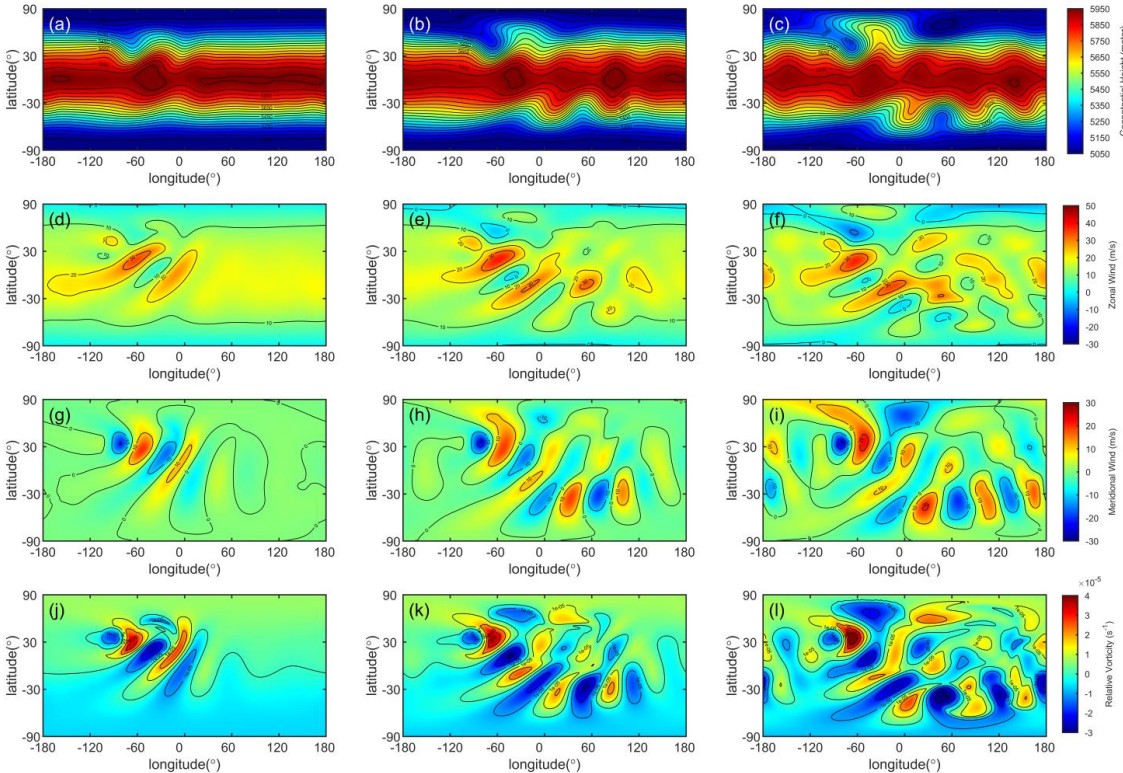


**Figure 8** Simulation result of mountain wave on C90 grid. The rows stand for variables: geopotential height, zonal wind, meridional
wind and relative vorticity, respectively. The columns represent simulation day 5, 10, 15. Geopotential height contour from 5050
to 5950 $m$ with interval 50 $m$. Zonal wind contour from $-30$ to 50 $m/s$ with interval 10 $m/s$. Meridional wind contour from $-30$
to 30 $m/s$ with interval 10 m/s. Relative vorticity contour from $-3 \times 10^{-5}$ to $4 \times 10^{-5}$ $s^{-1}$ with interval $1 \times 10^{-5}$ $s^{-1}$.

HOPE is able to deal with the bottom topography correctly, as shown in Figure 8, all of the simulation result is consistent

with prior researches such as (Nair et al., 2005a; Ullrich et al., 2010; Chen and Xiao, 2008) and so on. Furthermore, as
discussed in Bao et al. (2014), some high order Discontinuous Galerkin (DG) method exhibit non-physical oscillation during
simulating the over mountain flow, the additional viscosity operators are necessary to alleviate this issue. However, HOPE
does not require any explicit viscosity operator to suppress vorticity oscillations, the vorticity fields are smooth all the time
as illustrated in Figure 8 (j), (k), (l).





## 5.3 Rossby-Haurwitz Wave with 4 Waves


Rossby-Haurwitz (RH) wave is the 6[th] test case introduced by Williamson et al. (1992), the RH waves are analytic
solution of the spherical nonlinear barotropic vorticity equation, the reference solution is the zonal advection of RH wave
without pattern changing, the angular phase speed is given by

$$c = \frac{R(R+3)\omega - 2\Omega}{(R+1)(R+2)} \tag{69}$$

where $R = 4$ is the zonal wavenumber, $\omega = 7.848 \times 10^{-6} \, s^{-1}$; the earth rotation angular speed $\Omega = 7.292 \times 10^{-5} \, s^{-1}$.
Therefore, we have $c = 29.52 \, day$. The initial condition expressed as

$$\phi = \phi_0 + a^2[A(\theta) + B(\theta)\cos R\lambda + C(\theta)\cos 2R\lambda] \tag{70}$$

$$u = a\omega\cos\theta + aK\cos^{R-1}\theta \, (R\sin^2\theta - \cos^2\theta)\cos R\lambda \tag{71}$$

$$v = -aKR\cos^{R-1}\theta\sin\theta\sin R\lambda \tag{72}$$

$$A(\theta) = \frac{\omega}{2}(2\Omega + \omega)\cos^2\theta + \frac{1}{4}K^2\cos^{2R}\theta \, [(R+1)\cos^2\theta + 2R^2 - R - 2 - 2R^2\cos^{-2}\theta] \tag{73}$$

$$B(\theta) = \frac{2(\Omega + \omega)K}{(R+1)(R+2)}\cos^R\theta \, [R^2 + 2R + 2 - (R+1)^2\cos^2\theta] \tag{74}$$

$$C(\theta) = \frac{1}{4}K^2\cos^{2R}\theta \, [(R+1)\cos^2\theta - R - 2] \tag{75}$$

where $\lambda, \theta$ are longitude and latitude, $K = \omega, \phi_0 = gh_0, h_0 = 8000 \, m$, and $a = 6371220 \, m$ is the earth radius.
According to the study by Thuburn and Li (2000), the Rossby-Haurwitz (RH) wave with wavenumber 4 is unstable and
prone to waveform collapse due to factors such as grid symmetry, initial condition perturbation, and model errors. Similar
conclusions have been verified in subsequent research. In tests conducted by Zhou et al. (2020), the TRiSK framework based
on the SCVT grid could only sustain the RH wave pattern for 25 days without collapse. In contrast, (Li et al., 2020)
successfully maintained the RH wave pattern for 89 days using a similar algorithm on a latitude-longitude grid. Ullrich et al.
(2010) developed the high-accuracy finite volume model based on a cubed-sphere grid, which was able to sustain the RH
wave for up to 90 days. In the most of our experiments, the ability of HOPE to maintain the Rossby-Haurwitz (RH) wave
significantly improved with increased accuracy and grid resolution.
In the 3[rd] order accuracy simulation, we found that the duration for which the RH wave is maintained increases with
higher grid resolution, as exhibit in Figure 9. When the grid resolution is low (C45, C90), an obvious dissipation phenomenon
can be observed. When the resolution reaches C180, the dissipation is significantly reduced, but the waveform has completely
collapsed by day 90. When the resolution reaches C360, the simulation results are further improved, with dissipation further
reduced, and the RH wave waveform can still barely be maintained on day 90.



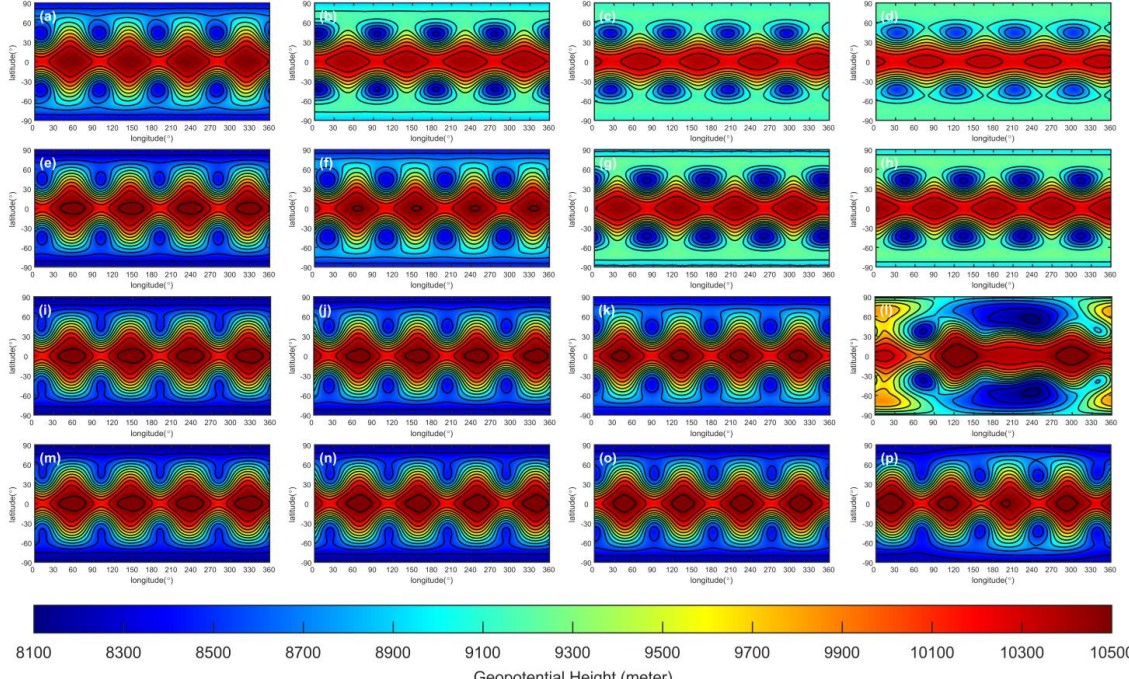

**Figure 9** Geopotential height of Rossby-Haurwitz wave simulated by 3$^{rd}$ order spatial reconstruction scheme. The rows represent grid C45, C90, C180 and C360, the columns stand for simulation day 14, 30, 60, 90. Contours from 8100 to 10500 $m$ with interval 200 $m$.

In Figure 10, we compare the impact of accuracy on the simulation capability of RH waves by fixing the resolution. By comparing row by row, it can be observed that when the accuracy reaches 5th order or higher, the dissipation is significantly reduced. Both the 5$^{th}$ order and 7$^{th}$ order accuracy simulations show signs of waveform distortion on day 90, and the waveform completely collapses by day 100. However, when using 9$^{th}$ order accuracy for the simulation, the waveform is well maintained even until day 100.



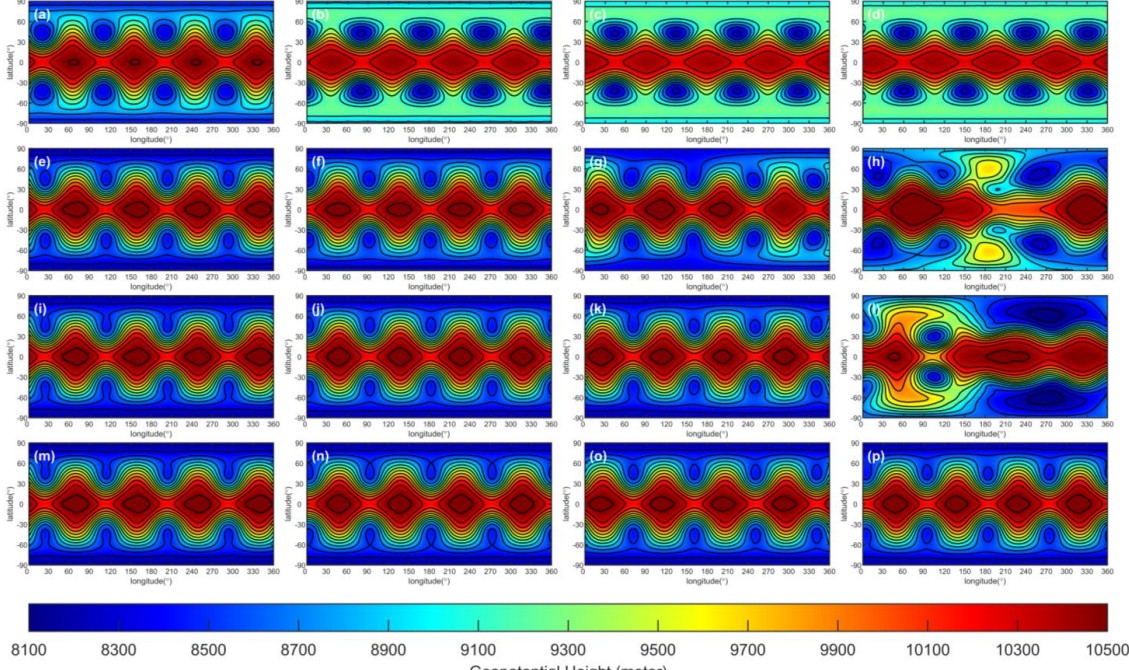

**Figure 10** Geopotential height of Rossby-Haurwitz wave on C90 grid, the rows represent the spatial reconstruction scheme with 3rd, 5th, 7th, 9th order, the columns stand for simulation day 30, 60, 90 and 100. Contours from 8100 to 10500 $m$ with interval 200 $m$.

Figure 11 presents the simulation results on the 80th day for different resolutions and accuracy schemes. The dissipation decreases as the resolution and accuracy improve. At the C45 resolution, both the 3rd order and 5th order accuracy simulations exhibit significant dissipation. Although the 7th order simulation shows a notable improvement in dissipation, the waveform is severely distorted. The 9th order accuracy scheme produces the best simulation results. As the resolution increases, the simulation performance also improves significantly. When using the C360 resolution, all accuracy schemes yield good simulation results.



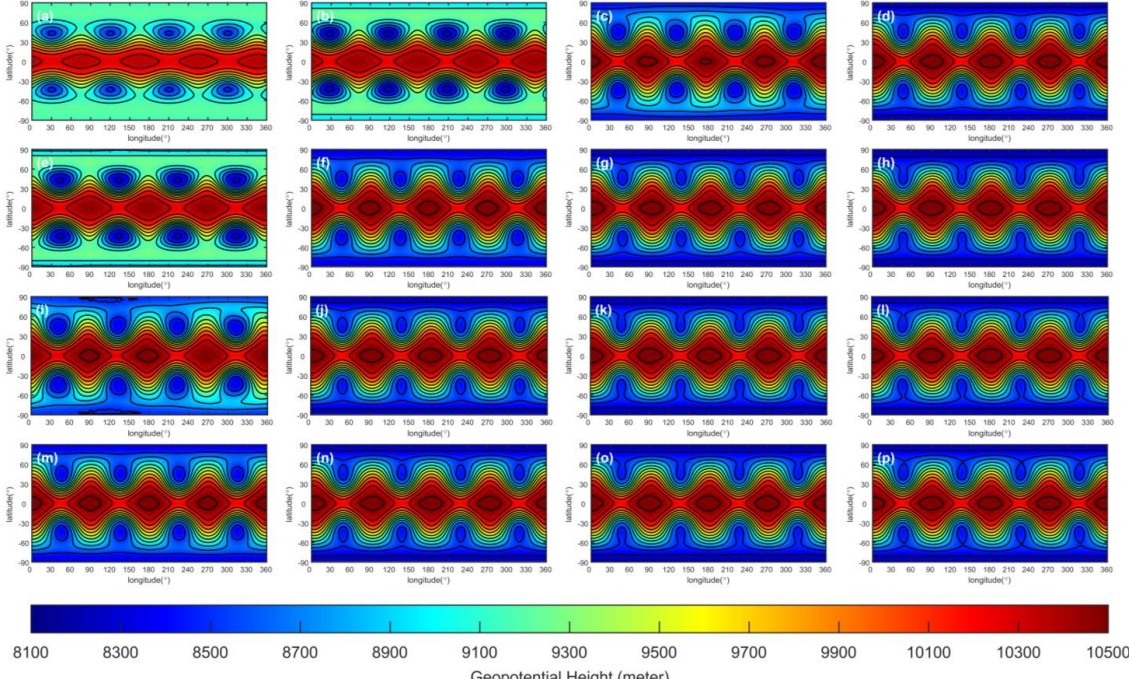

**Figure 11** Geopotential height of Rossby-Haurwitz wave at simulation day 80. The rows represent spatial reconstruction with 3rd,

5th, 7th and 9th order. The columns stand for grid C45, C90, C180 and C360. Contours from 8100 to 10500 $m$ with interval 200 $m$.

## 5.4 Perturbed Jet Flow

The perturbed jet flow was introduced by Galewsky et al. (2004), this experiment was desired to test the model ability

of simulating the fast and slow motion. the initial field is defined as

$$u(\theta) = \begin{cases} \dfrac{u_{max}}{e_n} e^{\frac{1}{(\theta-\theta_0)(\theta-\theta_1)}}, & \theta \in (\theta_0, \theta_1) \\ 0, & otherwise \end{cases} \tag{76}$$

$$\phi(\lambda, \theta) = \phi_0 + \phi'(\lambda, \theta) - \int_{-\frac{\pi}{2}}^{\theta} au(\theta') \left[ f + \frac{\tan \theta'}{a} u(\theta') \right] d\theta' \tag{77}$$

$$\phi'(\lambda, \theta) = g\hat{h} \cos \theta \, e^{-\left(\frac{\lambda}{\alpha}\right)^2 - \left(\frac{\theta_2 - \theta}{\beta}\right)^2}, \lambda \in (-\pi, \pi) \tag{78}$$

where $\lambda, \theta$ represents longitude and latitude, $a = 6371220 \, m$ is radius of earth, $u_{max} = 80 \, m/s, \theta_0 = \frac{\pi}{7}, \theta_1 = \frac{5\pi}{14}, \theta_2 = \frac{\pi}{4}$,

$e_n = e^{\frac{-4}{(\theta_1 - \theta_0)^2}}, \alpha = \frac{1}{3}, \beta = \frac{1}{15}$, and $\hat{h} = 120 \, m$.

As mentioned in Chen and Xiao (2008), the perturbed jet flow experiment poses a particular challenge for the cubed-

sphere grid model. Firstly, the jet stream is located at 45°N, which is very close to the boundaries of panel 5 of the cubed-

sphere grid, resulting in a large geopotential height gradient in the ghost interpolation region, which leads to larger





interpolation error. Furthermore, the location of the geopotential height perturbation $\phi'$ coincides with the boundary between
panel 1 and panel 5, which also leads to greater numerical computation errors.

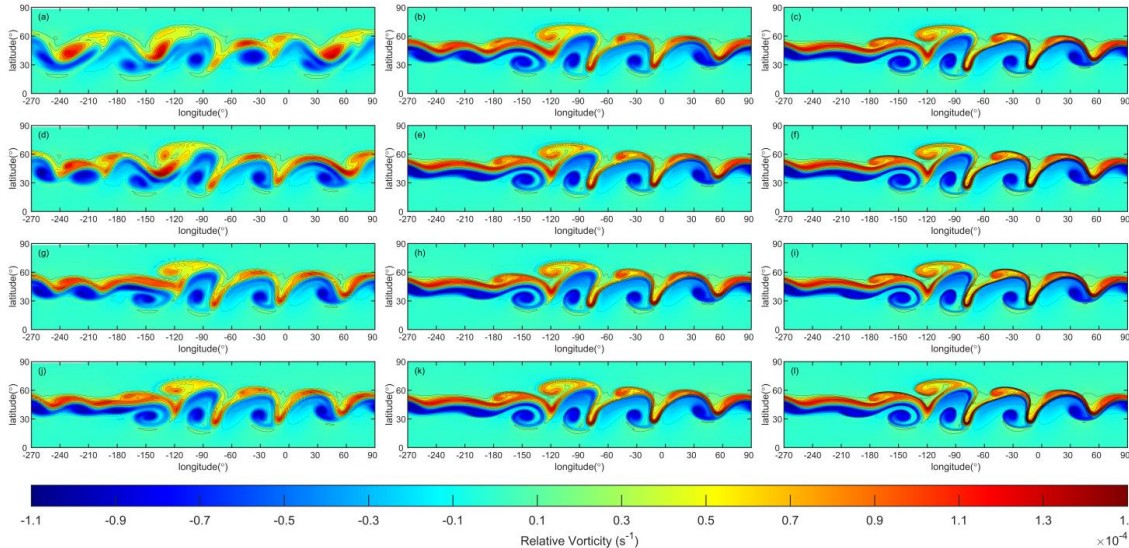


**Figure 12** Relative vorticity of perturbed jet flow. (a)~(c) represent the results of 5th order scheme with resolutions C45, C90, C180.
(d)~(f) represent the results of 7th order scheme with resolutions C45, C90, C180. (g)~(i) represent the results of 9th order scheme
with resolutions C45, C90, C180. (j)~(l) represent the results of 11th order scheme with resolutions C45, C90, C180.

Figure 12 displays the HOPE simulation outcomes at day 6 for varying levels of accuracy and resolutions. The four rows

correspond to the 5th, 7th, 9th, and 11th schemes in terms of accuracy. The three columns, meanwhile, represent the resolutions
of C45, C90, and C180, respectively. Upon comparing the different columns, it is evident that the perturbed jet flow test case
converges as the resolution increases. Figure 12 (a), (d), (g), and (j) illustrate that, with an increase in accuracy, the vorticity
field patterns become increasingly similar to the high-resolution results shown in the second and third columns of Figure 12.
Notably, HOPE enhances the simulation results by utilizing both higher accuracy and higher resolution.

## 5.5 Dam-Break Shock Wave

In this section we introduce a dam-break case for testing the capability of HOPE to capture the shock wave and comparing

the difference between 1D and 2D WENO schemes. The initial condition is configured as a cylinder with a height of 30000
meters, as shown in Figure 13(a). The geopotential height is given by

$$\phi(r(\lambda, \theta)) = \begin{cases} 2\phi_0, & r < r_c \\ \phi_0, & otherwise \end{cases} \tag{79}$$

where $r = \sqrt{(\lambda - \lambda_c)^2 + (\theta - \theta_c)^2}, \lambda_c = \pi, \theta_c = 0, r_c = \frac{\pi}{9}, \phi_0 = gh_0, h_0 = 30000\ m$, and the earth rotation angular speed
$\Omega = 0$.



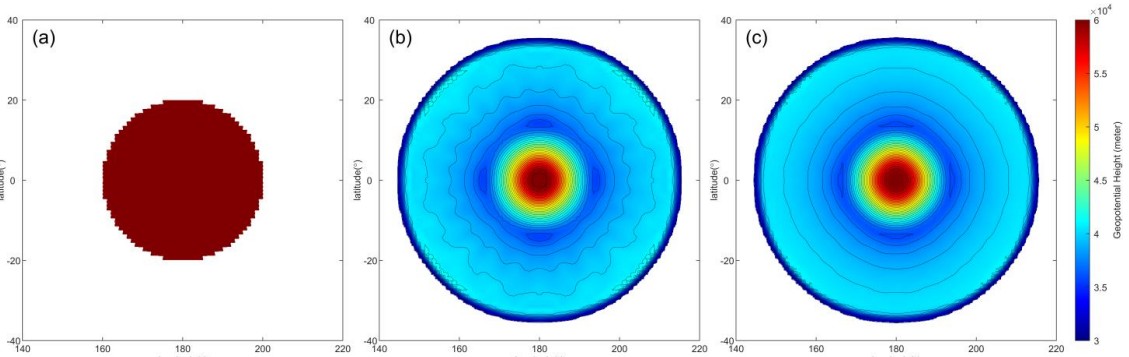

**Figure 13** Geopotential height of dam-break test case on C90 grid at 2nd hour. (a) Initial condition, (b) WENO 1D, (c) WENO 2D. The horizontal resolution for both schemes is C90. Shaded and contour from $3.2 \times 10^4$ to $6 \times 10^4$ meters, with contour interval $10^3$ meters.

In this experiment, we compare $5^{th}$ order accuracy on both 1D and 2D schemes, the WENO-Z (Borges et al., 2008) is adopted as WENO 1D scheme, and WENO 2D scheme is consist with section 3.2. Due to the initial condition being a cylinder, the resulting shock wave should maintain a circular feature. In the simulation results of WENO 1D, numerous radial textures appear, Figure 13(b). The simulation results using the WENO 2D scheme exhibit a smoother circular shape, Figure 13(c). This outcome arises because the 1D reconstruction scheme suffers from dimension split error, whereas the fitting function in the 2D reconstruction scheme incorporates cross terms, significantly improving the handling of anomalous anisotropic characteristics.

# 6. Conclusions

This paper presents HOPE, an innovative finite-volume model capable of achieving arbitrary odd-order convergence accuracy. Through comprehensive numerical experiments, we demonstrate that HOPE exhibits excellent convergence properties when applied to smooth flow fields, with simulation errors decreasing rapidly as the order of accuracy increases.

The model's performance has been rigorously evaluated across several benchmark cases:

1. In Rossby-Haurwitz wave simulations, HOPE demonstrates superior waveform preservation capabilities that scale with both spatial resolution and accuracy order.

2. For perturbed jet flow scenarios, the model successfully resolves both fast and slow dynamical features, with significant improvements in solution quality observed at higher orders and finer resolutions.

3. Mountain wave simulations confirm HOPE's ability to accurately represent orographically-forced gravity waves.

4. In the dam break test case featuring cylindrical shock fronts, the two-dimensional WENO reconstruction scheme proves more effective than dimension-split approaches in maintaining circular symmetry.



A key innovation of HOPE lies in its computational architecture. The algorithm is specifically designed to harness GPU
acceleration through (1) Implementation of spatial reconstructions as convolutional operations, and (2) Formulation of
integration steps as matrix-vector products. These design choices leverage computational patterns widely adopted in machine
learning frameworks. By developing HOPE within PyTorch, we inherit automatic differentiation capabilities, enabling
straightforward coupling with neural network systems.
This integration facilitates the development of hybrid prediction models that combine a high-order, high-performance
dynamical core, and Neural network-based physical parameterizations. Current research efforts have successfully extended
this algorithmic framework to a two-dimensional baroclinic model (X-Z dimensions).
Future work will focus on developing a global, fully compressible baroclinic model using the HOPE algorithm, further
demonstrating its versatility and advantages for modeling complex atmospheric dynamics. The model's unique combination
of physical conservation, computational efficiency, and machine learning compatibility positions it as a powerful tool for
next-generation atmospheric modeling.

# 7. Appendix

In this appendix, we introduce a novel boundary ghost cell interpolation scheme for cubed sphere, which is able to
support HOPE to reach the accuracy over $11^{th}$ order or even higher.
There are two types of cells, in-domain and out-domain (also named ghost cell, as show in Figure 7(b)), we define the
set of in-domain cell values $\boldsymbol{q}_{d\times1} = (q_1, q_2, ..., q_d)^T$, the set of out-domain cell values $\boldsymbol{g}_{h\times1} = (g_1, g_2, ..., g_d)^T$, and the set
of Gaussian quadrature point values (green points in Figure 3) in out-domain cells is define as $\boldsymbol{v}_{p\times1} = (v_1, v_2, ..., v_p)$. To
identify the shape of the arrays, we denote the array shape using subscripts (this convention will be followed throughout the
subsequent text). The purpose of ghost cell interpolation is using the known cell value $\boldsymbol{q}$ to interpolate the unknown $\boldsymbol{g}$.
Define a new set includes the values of domain cell values and ghost cell values

$$\widetilde{\boldsymbol{q}}_{(d+h)\times1} = \boldsymbol{q} \cup \boldsymbol{g} = (q_1, q_2, ..., q_d, g_1, g_2, ..., g_h)^T$$

(A.1)

Similar to the describe in section 3.1, we can use a TPP to reconstruct the ghost quadrature points

$$\boldsymbol{v}_{p\times1} = A_{p\times(d+h)}\widetilde{\boldsymbol{q}}_{(d+h)\times1}$$

(A.2)

where $A_{p\times(d+h)}$ is the interpolation matrix that can be obtain by the similar method to Eq.(27). The ghost cell values are
calculated by Gaussian quadrature

$$\boldsymbol{g}_{h\times1} = B_{h\times p}\boldsymbol{v}_{p\times1}$$

(A.3)

where $B_{h\times p}$ is the Gaussian quadrature matrix.
$\widetilde{\boldsymbol{q}}_{(d+h)\times1}$ can be decomposed as the linear combination of $\boldsymbol{q}_{d\times1}$ and $\boldsymbol{v}_{p\times1}$



$$\widetilde{\boldsymbol{q}}_{(d+h)\times 1} = \begin{pmatrix} I_{d\times d} & 0 \\ 0 & B_{h\times p} \end{pmatrix} \begin{pmatrix} \boldsymbol{q}_{d\times 1} \\ \boldsymbol{v}_{p\times 1} \end{pmatrix} = \tilde{B}_{(d+h)\times(d+p)}\overline{\boldsymbol{q}}_{(d+p)\times 1} \tag{A.4}$$

where $I_{d\times d}$ is an identity matrix, and

$$\tilde{B}_{(d+h)\times(d+p)} = \begin{pmatrix} I_{d\times d} & 0 \\ 0 & B_{h\times p} \end{pmatrix} \tag{A.5}$$

$$\overline{\boldsymbol{q}}_{(d+p)\times 1} = \begin{pmatrix} \boldsymbol{q}_{d\times 1} \\ \boldsymbol{v}_{p\times 1} \end{pmatrix} \tag{A.6}$$

Substitute Eq.(28) into Eq.(24), we have

$$\boldsymbol{v}_{p\times 1} = A_{p\times(d+h)}\tilde{B}_{(d+h)\times(d+p)}\overline{\boldsymbol{q}}_{(d+p)\times 1} = \tilde{A}_{p\times(d+p)}\overline{\boldsymbol{q}}_{(d+p)\times 1} = \tilde{A}_{p\times(d+p)} \begin{pmatrix} \boldsymbol{q}_{d\times 1} \\ \boldsymbol{v}_{p\times 1} \end{pmatrix} \tag{A.7}$$

We found that matrix $\tilde{A}_{p\times(d+p)}$ can be decomposed into two parts

$$\tilde{A}_{p\times(d+p)} = \begin{pmatrix} \bar{A}_{p\times d} & C_{p\times p} \end{pmatrix} \tag{A.8}$$

Such that

$$\boldsymbol{v}_{p\times 1} = \bar{A}_{p\times d}\boldsymbol{q}_{d\times 1} + C_{p\times p}\boldsymbol{v}_{p\times 1} \tag{A.9}$$

Therefore

$$\left(I_{p\times p} - C_{p\times p}\right)\boldsymbol{v}_{p\times 1} = \bar{A}_{p\times d}\boldsymbol{q}_{d\times 1} \tag{A.10}$$

We set $D_{p\times p} = I_{p\times p} - C_{p\times p}$, then $\boldsymbol{v}_{p\times 1}$ can be determined by

$$\boldsymbol{v}_{p\times 1} = D_{p\times p}^{-1}\bar{A}_{p\times d}\boldsymbol{q}_{d\times 1} \tag{A.11}$$

Substitute Eq.(A.11) into Eq.(A.3), we establish the relationship between ghost cell values and in-domain cell values

$$\boldsymbol{g}_{h\times 1} = B_{h\times p}\boldsymbol{v}_{p\times 1} = B_{h\times p}D_{p\times p}^{-1}\bar{A}_{p\times d}\boldsymbol{q}_{d\times 1} = G_{h\times d}\boldsymbol{q}_{d\times 1} \tag{A.12}$$

where $G_{h\times d} = B_{h\times p}D_{p\times p}^{-1}\bar{A}_{p\times d}$. It's clear that Eq.(A.12) is linear, and only rely on the mesh and Gaussian quadrature scheme. Therefore, we need to compute the projection matrix $G_{h\times d}$ only once for a given mesh and accuracy, this matrix can be computed by a preprocessing system and save it to the hard disk.

*Code availability.* The digital object identifier (DOI) for HOPE shallow water model is 10.5281/zenodo.15351234 (Zhou, 2025).

*Author contributions.* LZ developed the algorithm, implemented the code, conducted numerical experiments, and wrote the manuscript. WX provided expert guidance on software architecture design and high-performance computing implementations.

*Competing interests.* The authors declare that they have no conflict of interest.

*Financial support.* This work is partially supported by National Natural Science foundation of China (No. U2242210)



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
