# Peer review of "HOPE: An Arbitrary-Order Non-Oscillatory Finite-Volume Shallow"

_EGUsphere, 2025_

## Author Comment (AC1)

Upon reviewing your comments, I recognize that you have conducted a detailed and rigorous examination of the manuscript, even cross-referencing it with the source code. I am deeply impressed by such dedication to scientific rigor. Before addressing your points individually, I would first like to express my sincere appreciation for your meticulous scientific approach, exceptional expertise, and profound theoretical insight. The numerous comments you provided are invaluable for enhancing the quality of this paper.

*Points related to interpretation and understanding*

**Referee Comment 1**

*Line 74. It is important to make sure that you compare like with like. A k'th order 1D finite difference derivative requires (generally) a stencil of k+1 points or cells. In HOPE you mostly discuss reconstructions rather than derivatives; a k'th order reconstruction can be done with a stencil of k cells. But if you take a difference of two reconstructions to compute a derivative then you will have used two different stencils and at least k+1 data points.*

**Response to Referee:**

Your point is entirely correct. In HOPE, we employ the finite volume method to discretize the equations. The increment of the cell-averaged prognostic variables stems from the net flux and source terms. Since a cell boundary is shared by two adjacent cells, the calculation of the flux utilizes reconstructed values from both sides. Therefore, to complete one Runge-Kutta substep integration for a given cell, information from a total of $(k + 2) \times (k + 2)$ cells is required. This is consistent with our statement on Line 74 that a reconstruction stencil of width k enables the construction of a $k^{th}$ order accurate scheme.

Furthermore, regarding comparisons: Ullrich and Jablonowski (2012, JCP, Section 4.2) constructed a class of 4th-order accurate schemes. They first expressed derivatives using a rhombic stencil spanning 5 cells in width. These derivatives were then used to reconstruct the state at cell interfaces for input to a Riemann solver. Consequently, completing one Runge-Kutta substep integration for a cell required information from neighbor cells within a rhombic region spanning 7 cells in width.

Within the HOPE framework, a rhombic stencil can also be employed. When configured as described above, HOPE achieves 5th-order accuracy with such a stencil, bypassing the need for

complex derivative relationships. However, it is noteworthy that when using a rhombic stencil for reconstruction, the stability of explicit time integration is adversely affected. Compared to the rectangular stencils used in the present work, rhombic stencils necessitate a shorter time step to maintain computational stability. Although rhombic stencils slightly reduce reconstruction computational cost, this benefit does not compensate for the efficiency loss due to the reduced time step. Furthermore, in implementation, reconstruction using rectangular stencils can be efficiently mapped to PyTorch's conv2d function, offering advantages in both computational efficiency and implementation simplicity.

**Referee Comment 2**

*High order*

- *It is good to see the requirement for smoothness of data mentioned for high order to be more accurate (line 65, 81-82). Advocates of high order schemes don't always mention this.*

- *However, it is important to be precise with terminology to avoid confusion for readers (and authors!) The phrase 'convergence accuracy' (line 69) mixes up two ideas that should be kept distinct: order of accuracy and convergence rate. Convergence rate agrees with order of accuracy only for sufficiently smooth data. It is very common for the convergence rate to be less than the order of accuracy.*

- *Line 77. 'arbitrary accuracy' -> arbitrary order of accuracy. Check for other places where you have used 'accuracy' when you mean 'order of accuracy' (e.g., lines 424, 433, 458, 461, 463). Order of accuracy is not the same thing as accuracy; there are even situations where a higher order of accuracy produces a less accurate solution.*

**Response to Referee:**

Your description aligns with our understanding: accuracy refers to the closeness of the numerical solution to the true value, while reconstruction order characterizes the property of the numerical scheme. For high-order schemes, the convergence rate achieves the designed reconstruction order only when the flow field is sufficiently smooth. We have thoroughly reviewed and revised the corresponding sections of the manuscript to clearly distinguish between the terms accuracy, reconstruction order, and convergence rate.

**Referee Comment 3**

*Line 224. It is not obvious that negative values of (an element of) \gamma could cause instability. Presumably the elements of R_H must be allowed to be negative, otherwise we would not be able to achieve more than second order? So why should there be such a restriction on \gamma? Please give some discussion or a reference.*

**Response to Referee:**

The conclusion that negative $\gamma$ values cause computational breakdown originates from the study by Shi et al. 2002, entitled "A Technique of Treating Negative Weights in WENO Schemes" (already cited in the manuscript).

The following description is cited from Shi et al. 2002 Section 1:

*We remark that negative linear weights do not appear in finite difference WENO schemes in any spatial dimensions for conservation laws for any order of accuracy…, Unfortunately, they do appear in some other cases, such as the central WENO schemes using staggered meshes we have seen above, high-order finite volume schemes for two dimensions described in [8] and in this paper, … In fact, in all the test cases involving negative linear weights and discontinuous solutions presented in this paper, WENO schemes without special treatment to the negative weights are unstable (the numerical solution blows up and the code stops).*

**Referee Comment 4**

*At first glance (41) seems to be dimensionally inconsistent, since it mixes derivatives of different orders. It might be good to remind the reader that the computational coordinates x and y have effectively been non-dimensionalized by the grid spacing so that \Delta x = \Delta y = 1. (Thus, (41) is dimensionally correct, after all.) This non-dimensionalization is also appropriate to ensure that the smoothness indicators \beta_i scale with resolution in an appropriate way. What is \epsilon in (39)?*

**Response to Referee:**

We agree with your suggestion and have implemented the following revisions to the relevant portion in Section 3.2:

$$\alpha_i^\pm = \gamma_i^\pm \left( 1 + \frac{\tau}{\beta_i + \varepsilon} \right)$$

where $\varepsilon = 10^{-14}$ is introduced to prevent division by zero. The smooth indicators $\beta_i$ quantify the smoothness of reconstruction functions within the target cell. We employ a formulation analogous to that described in,

As mentioned in Section 错误!未找到引用源。, all cells participating in reconstruction within HOPE's computational space can be treated as identical unit squares with $\Delta x = \Delta y = 1$. Thus, the smooth indicators are expressed as:

$$\beta_i = \sum_{\zeta=1}^{l} \iint_\Omega \left[ \frac{\partial^\zeta}{\partial x^{\zeta_1} \partial y^{\zeta_2}} p_j(x, y) \right]^2 dxdy$$

where $\zeta_1 + \zeta_2 = \zeta$ and $\zeta > 0$, $\zeta_1, \zeta_2 \in [0, n]$, and $l$ is the sub-stencil width.

**Referee Comment 5**

*Section 5, general comment (to the whole community, really!): make sure you extract good, useful information from your test cases, not just attractive-looking plots! For example, see the next point, as well as the suggestion below to diagnose dissipation quantitatively.*

*Some of the test cases in section 5 are run with the high-order reconstruction and some are run with the WENO scheme, and the flow over the mountain case does not say which scheme is used. In an operational model one must make up one's mind which scheme to use, though, in research mode, having different options available allows one to explore sensitivities. It would be valuable for readers if you could share any knowledge and understanding you have gleaned by comparing WENO vs high order on the different test cases. Even if you don't show figures and tables for all combinations, it would be good to comment on any differences. For example, do WENO3 and WENO5 give 3rd order and 5th order convergence for the steady geostrophic flow case? Do high order schemes produce oscillations in the flow over a mountain case? Does WENO3 produce similar solutions to the third order scheme on the Rossby-Haurwitz test case?*

**Response to Referee:**

Thank you for your valuable suggestions and questions. In response, we have clarified the specific schemes used in each test case within the revised manuscript and have included new experimental results.

1. **Steady Geostrophic Flow:** Both WENO3 and WENO5 achieve the expected 3rd-order and 5th-order convergence rates, respectively. However, the computed norm errors for WENO schemes are marginally larger than those obtained with the TPP3 and TPP5 schemes. This observation confirms that the 2D WENO scheme preserves the designed convergence rate in smooth flow regions. Concurrently, in the Dam-Break Shock Wave case, the 2D WENO scheme demonstrates its robust capability for handling discontinuous flow fields. These combined results align perfectly with the primary motivation for introducing the WENO scheme: its adaptive oscillation suppression capability. Specifically, the scheme preserves the high convergence rate in sufficiently smooth regions while automatically reducing the reconstruction order near discontinuities to effectively suppress the development and propagation of non-physical oscillations.

2. **Rossby-Haurwitz Wave**: Significant differences were observed between the 2D WENO scheme and the TPP schemes in this test. Regardless of the specific WENO order employed (3, 5, 7, or 9), all WENO variants maintained the Rossby-Haurwitz (RH) wave pattern for a shorter duration compared to their TPP counterparts of equivalent order. We infer that the nonlinear processes inherent within the WENO scheme introduce asymmetries that disrupt the computational stencil symmetry, leading to a premature breakdown of the RH wave.

3. **Zonal Flow over an Isolated Mountain**: We tested TPP3, TPP5, TPP7, WENO3, and WENO5 schemes for this case. The simulation results from all these schemes were found to be closely matched.

To explicitly state these conclusion, we have added clarifying remarks in the manuscript.

**Referee Comment 6**

*Line 407. 'prone to collapse due to factors such as...'. To be clear, the R=4 Rossby-Haurwitz wave is unstable. Those factors, or even roundoff error, can provide a perturbation that initiates the instability, but they are not the fundamental cause of the collapse - that is the instability itself.*

**Response to Referee:**

We appreciate your identification of the inaccuracy in our manuscript's description. We confirm that the fundamental cause of waveform collapse in the Rossby-Haurwitz wavenumber-4 solution is its intrinsic dynamical instability—not external factors such as grid symmetry, initial

perturbations, or model errors. The role of these external factors is to act as sources of minute perturbations that may *trigger* this inherent instability or influence the timing and manner of its manifestation in numerical simulations. Accordingly, we have revised the relevant text as follows:

*"According to the study by Thuburn and Li (2000), the Rossby-Haurwitz (RH) wave with wavenumber 4 is inherently dynamically unstable and prone to collapse. This instability can be triggered by minute perturbations, such as those arising from grid structure (breaking initial symmetry), initial condition imperfections, or numerical errors (e.g., truncation or roundoff)."*

**Referee Comment 7**

*Line 491. That test case is actually dominated by Rossby waves, not gravity waves.*

**Response to Referee:**

We appreciate this correction and have revised the manuscript accordingly.

**Referee Comment 8**

***Points for discussion; the authors may or may not wish to address these in the manuscript.***

*Riemann solver*

- *The Riemann solver is applied at every quadrature point before doing the quadrature. Would there be any advantage in doing the quadrature first and then applying the Riemann solver just once at each interface?*

**Response to Referee:**

The sequence of quadrature and Riemann solver operations is not arbitrary. In the finite volume method, the conservation law is formulated as:

$$\frac{\partial \overline{q}_{i,j}}{\partial t} + \frac{\overline{F}_{i+\frac{1}{2},j} - \overline{F}_{i-\frac{1}{2},j}}{\Delta x} + \frac{\overline{G}_{i,j+\frac{1}{2}} - \overline{G}_{i,j-\frac{1}{2}}}{\Delta y} = \overline{S}_{i,j}$$

Consequently, computing the temporal tendency of the cell-averaged solution requires the integral averages of fluxes $F$ and $G$ along cell interfaces. Considering only the x-direction flux, the interface-averaged flux is defined as:

$$\overline{F}_{i+\frac{1}{2},j} = \frac{1}{\Delta y} \int_{e_{i+\frac{1}{2},j}} F(q)\, dy$$

This requires first computing $\boldsymbol{F}$ from $\boldsymbol{q}$ followed by integration. Conversely, if one first computes the interface-averaged state:

$$\overline{\boldsymbol{q}}_{i+\frac{1}{2},j} = \frac{1}{\Delta y} \int_{e_{i+\frac{1}{2},j}} \boldsymbol{q}\, dy$$

and then applies the Riemann solver, the resulting flux becomes:

$$\widetilde{\boldsymbol{F}}_{i+\frac{1}{2},j} = \boldsymbol{F}\left(\overline{\boldsymbol{q}}_{i+\frac{1}{2},j}\right) = \boldsymbol{F}\left(\frac{1}{\Delta y} \int_{e_{i+\frac{1}{2},j}} \boldsymbol{q}\, dy\right)$$

Due to the nonlinear nature of $\boldsymbol{F}$:

$$\overline{\boldsymbol{F}}_{i+\frac{1}{2},j} \neq \widetilde{\boldsymbol{F}}_{i+\frac{1}{2},j}$$

Numerical testing confirms that using $\widetilde{\boldsymbol{F}}_{i+\frac{1}{2},j}$ as the flux limits accuracy to second-order regardless of reconstruction scheme order.

Furthermore, in curved coordinate systems where metric tensors vary spatially. The averaged quantity $\overline{\boldsymbol{q}}_{i+\frac{1}{2},j}$ lacks physical correspondence to any specific point on the interface. This ambiguity prevents determination of both: (i) The spatial location represented by $\overline{\boldsymbol{q}}_{i+\frac{1}{2},j}$; (ii) The appropriate metric tensor position for flux computation. Thus invalidating accurate calculation of $\overline{\boldsymbol{F}}_{i+\frac{1}{2},j}$.

**Referee Comment 9**

*High order*

- *Line 378. Do you have any data on how much more expensive high order is? Of course the answer will depend on implementation, computing platform, resolution, etc, but it would be good to have an idea.*

**Response to Referee:**

We conducted 15-day simulations for the "Zonal Flow Over an Isolated Mountain" test case using TPP5 and TPP7 schemes at C180 resolution. The time integration step was 100 seconds, executed on an NVIDIA RTX 3090 GPU. Computational costs were measured as follows:

 **TPP5**:

32-bit precision: 144 s

64-bit precision: 469 s

**TPP7**:

32-bit precision: 160 s

64-bit precision: 730 s

The NVIDIA GeForce RTX 3090 GPU is primarily designed for single-precision (FP32) compute workloads, lacking dedicated hardware for high-performance double-precision (FP64) computation. Its FP64 performance is significantly lower, typically operating at 1/64th of its peak FP32 throughput (according to https://www.techpowerup.com/gpu-specs/geforce-rtx-3090.c3622). Consequently, FP64 execution times increase substantially with scheme order, whereas the computational cost difference between TPP5 and TPP7 under FP32 is marginal at 11%.

- *Line 380. I am inclined to agree that 3rd or 5th order will be the practical choice, but I would be interested to know your reasoning.*

**Response to Referee:**

Our preference for 3rd/5th-order schemes stems from empirical observations in HOPE's baroclinic version and studies of other high-order models (e.g., Multi-moment Constrained finite Volume model developed by CMA Earth System Modeling and Prediction Centre). We note diminishing returns in forecast accuracy improvement beyond these orders due to three fundamental constraints:

1. **Dominant Parameterization Errors**

   Forecast errors originate from dynamical cores, physical parameterization schemes, and data assimilation. Current subgrid physical processes remain poorly characterized, with parameterizations relying heavily on empirical functions. These parameterization-induced errors dominate total forecast error, rendering excessive dynamical core refinement ineffective for error reduction.

2. **Resolution-Precision Interdependence**

   a) Smoothness Requirement: High-order schemes achieve theoretical convergence rates only for sufficiently smooth flows. Atmospheric discontinuities (e.g., planet boundary layer heating, turbulence, orography) frequently violate this condition.

b) Wave Resolution Limits: Discrete systems require ≥2 gridpoints per wave - even with infinite-order schemes. Gridpoint models accurately resolve only wavelengths >4-6Δx.

c) Scale Separation: Increasing reconstruction order without grid refinement cannot improve subgrid-scale representation; enhanced resolution remains essential for smaller-scale systems.

3. **Computational Efficiency**

Extensive HOPE testing reveals:

a) 5th-order schemes provide substantially better accuracy than 3rd-order

b) Higher orders yield diminishing accuracy gains disproportionate to computational cost increases

**Conclusion**: While higher-order schemes improve smooth-flow representation, their computational expense and resolution constraints make 5th-order reconstruction the optimal balance between precision and cost-effectiveness in current atmospheric modeling.

We emphasize the exploratory nature of this perspective. Given rapid advancements in artificial intelligence, whether integrating high-order dynamical cores with AI-based physical parameterization schemes will yield significant advances remains an open research question. We would welcome opportunities for deeper discussions with experts of your caliber on this evolving frontier.

**Referee Comment 11**

- *One very useful potential application of a code like HOPE would be to help answer that question in a quantitative way. Flows of realistic complexity (therefore not very smooth), like the flow over an isolated mountain case, generally don't have exact solutions available. But if you could compute an accurate high-resolution reference solution then you would be able to plot error versus computational cost as you vary both resolution and order of accuracy.*

**Response to Referee:**

We appreciate the constructive suggestion. Traditionally, such analyses have relied on spectral models: a relatively accurate solution is first obtained via spectral methods to establish a benchmark for evaluating other models' simulation capabilities. Your proposed quantitative

framework provides valuable inspiration for HOPE's development, and we will incorporate such analyses in future work.

Based on our current assessment of HOPE's performance, for smooth flow fields, increasing the order of accuracy yields significantly greater benefits than refining spatial resolution. This is because higher resolution necessitates reduced time steps in explicit time integration. We concur that your approach is particularly well-suited for analyzing the trade-off between simulation accuracy versus computational cost under complex topographic conditions – specifically, identifying optimal combinations of resolution and accuracy order that achieve sufficiently accurate results while balancing computational expense and error magnitude.

**Referee Comment 12**

*Rossby-Haurwitz wave collapse*

- *Lines 414-418. How long the Rossby-Haurwitz wave is sustained is a measure of how strongly numerical errors project onto the growing mode(s) at early times. After that the instability grows at its own rate until the RH wave collapses.  Note that a cubed sphere (in the usual orientation) has an advantage (compared to an icosahedral grid, for example) in that its discretization errors should project onto zonal wavenumber 4 and higher harmonics, whereas zonal wavenumbers 1, 3, and 5 project onto the instability. Thus, presumably, it is roundoff errors that break the wavenumber 4 symmetry and trigger the instability for HOPE(?) If that is the case, then higher precision should delay the collapse. Have you tested that? It sounds like you are set up to be able to do that easily. Conversely, if higher precision does not delay the collapse, then that begs the question: what is breaking the wavenumber 4 symmetry to trigger the instability, and could it be an implementation bug?*

**Response to Referee:**

Your assessment is correct. The description in our manuscript (lines 424-428) intended to convey the same point as your comment here. This also made us realize the need for a clearer and more detailed explanation of this conclusion.

Figure 10 in the manuscript shows simulation results using a C90 grid. The four rows correspond to simulations using the TPP3, TPP5, TPP7 and TPP9 schemes, respectively.

The four columns represent simulation durations of 30, 60, 90, and 100 days. For example, Figure 10(h) is located in the second row (5th-order scheme) and the fourth column (100 days).

[Figure]

**Figure 1** Geopotential height of Rossby-Haurwitz wave on C90 grid, the rows represent the spatial reconstruction scheme with TPP3, TPP5, TPP7 and TPP9, the columns stand for simulation day 30, 60, 90 and 100. Contours from 8100 to 10500 $m$ with interval 200 $m$.

Specifically:

1. The first row of subfigures TPP3 (3rd-order scheme) shows that although the wave pattern remains intact (does not collapse) as simulation time increases, the maximum geopotential height exhibits a significant decrease compared to the initial state.

2. For the TPP5 scheme (second row), the wave becomes distorted by day 90 and completely collapses by day 100 (Figure 10(h)).

3. Using the TPP7 scheme (third row), wave distortion at day 90 shows some improvement, but the pattern still cannot be maintained until day 100.

4. When simulating the RH wave with the TPP9 scheme (fourth row), the maximum geopotential height is well preserved, and the wave pattern shows no collapse even at day 100.

Based on this analysis, we can conclude that for the HOPE model, employing a higher-order scheme indeed enhances the ability to maintain the RH wave structure, i.e., delays the collapse.

**Referee Comment 13**

- *Also, it is good to be aware of what the time of collapse is really telling you about the model formulation, and to look at other informative aspects of the solution. For example, you mention apparent 'dissipation' of the solution. You could measure that dissipation quantitiatively by diagnosing a conserved quantity like energy or potential enstrophy, for example, and look at how their conservation depends on resolution and order of accuracy.*

**Response to Referee:**

Your suggestion is very well-founded. Therefore, we have incorporated a new figure into the manuscript depicting the evolution of total energy and maximum geopotential height over time. The following addition has been made to the manuscript:

We define the relative error $\epsilon_r$ of the variable $q$ as $\epsilon_r = \frac{q^n - q^0}{q^0}$, where $q^0$ and $q^n$ stand for $q$ value at initial time and time slot $n$, respectively. A 100-day simulation of the Rossby-Haurwitz wave was conducted using a C90 grid (1° resolution). For the fifth-, seventh-, and ninth-order schemes, the total energy maintained a dynamic balance. The relative error in total energy oscillated regularly within a range of $-4.6 \times 10^{-3}$ to 0. Furthermore, the variation patterns of this relative error were remarkably similar across these three schemes. In contrast, the total energy evolution of the third-order scheme differed significantly from the others. After 30 days of simulation, the third-order scheme exhibited anti-dissipation in total energy, reaching a maximum relative error of $6.6 \times 10^{-4}$, as illustrated in Figure 2 (a).

However, this greater energy retention did not translate to a better ability of the third-order scheme to maintain the maximum geopotential height. As shown in Figure 2 (b), the fifth-, seventh-, and ninth-order schemes demonstrated a comparable ability to maintain the maximum geopotential height, exhibiting significant anomalies only after the Rossby-Haurwitz waveform collapsed. In the simulation using the third-order scheme, the maximum geopotential height began to decrease notably starting from day 10.

[Figure]

**Figure 2** Evolution of relative errors for the Rossby-Haurwitz wave simulation on the C90 grid over days 0 to 100. (a) Relative error in total energy. (b) Relative error in maximum geopotential height. Line styles: red solid: 3rd-order scheme; blue dashed: 5th-order scheme; magenta solid: 7th-order scheme; black dashed: 9th-order scheme.

*Points related to improving the clarity of the explanations*

**Referee Comment 14**

*Line 18. '...reduces interpolation to matrix-vector multiplication'. When I first read this I thought it was stating the obvious: interpolation is a linear operation. The significance only became clear when I read section 3.3: even though the panel boundary treatment couples ghost points on the two sides of the boundary, it can be reduced to a straightforward matrix-vector multiplication. Perhaps you can briefly mention this two-way coupling in the abstract.*

**Response to Referee:**

Your understanding of our novel ghost cell interpolation scheme is precisely captured. We concur that the descriptor "two-way coupling" accurately reflects the algorithm's characteristics.

Accordingly, we have included a description of the two-way coupled ghost cell interpolation scheme in the abstract.

**Referee Comment 15**

*Abstract line 23. It is unclear why a separate fortran code version is needed for `convergence analysis'. The fortran version is not mentioned in the main text (though the source code is provided).*

**Response to Referee:**

As demonstrated in the Steady State Geostrophic Flow test, when using high-order schemes, HOPE achieves extremely small errors in simulating smooth flow fields even on very coarse resolutions. These errors can be so minute that they fall below the 16 significant digits representable in double precision. Under these conditions, conducting precision tests using double precision alone fails to accurately capture the true convergence rate. To obtain correct error measurements and convergence rate, we must employ FP128 (real(16) in Fortran). However, PyTorch's underlying architecture is built on NVIDIA CUDA, which currently supports only up to FP64 (double precision). Consequently, the PyTorch implementation cannot provide correct simulation errors when utilizing ultra-high-order schemes. We appreciate your observation and have added a description addressing this issue in the Steady State Geostrophic Flow Experiment section to clarify the implementation details.

**Referee Comment 16**

*Line 44. Comment: whether a spectral method conserves mass depends on which variables are chosen to be represented by a spectral expansion. For example, predicting a spectral representation of surface pressure (rather than the more usual log of surface pressure) should conserve mass in a hydrostatic model.*

**Response to Referee:**

It is true that spectral methods can achieve mass conservation with supplementary adjustments. However, our emphasis here is that finite volume methods inherently ensure strict mass conservation. This property stems from the fundamental principle of the finite volume approach: the change in the cell-averaged quantity equals the net flux across the cell boundary.

Consequently, strict mass conservation is guaranteed without requiring additional algorithmic design. Following your suggestion, we have revised this section to state:

*"While the implementation of a spectral dynamical core in NeuralGCM theoretically enables infinite-order accuracy, the global nature of spectral expansion restricts the method's scalability. Furthermore, in contrast to finite-volume algorithms which inherently ensure strict mass conservation, achieving strict mass conservation with NeuralGCM's spectral dynamical core requires supplementary modifications."*

**Referee Comment 17**

*Line 60. At this point it is unclear which Jacobian matrix you mean. Which derivatives are computed automatically? Some more explanation is needed. Similarly on line 320: which gradients can be computed efficiently?*

**Response to Referee:**

We appreciate this clarification. The original reference to the "Jacobian matrix" was indeed ambiguous. The matrix in question is specifically the ghost cell interpolation matrix. This terminology was adopted because its computational generation method closely resembles the algorithm used by PyTorch to compute Jacobian matrices. We have revised the text to consistently use ghost cell interpolation matrix for clarity.

Notably, our implementation for generating this matrix achieves significant acceleration and substantially reduces VRAM demand compared to PyTorch's native "torch.autograd.functional.jacobian" function. The key optimizations are:

1. Parallel Multi-Row Computation: Utilizing "torch.vmap" to encapsulate "torch.func.vjp", enabling simultaneous computation of multiple matrix rows.

2. CSR Compression & Incremental Disk Storage:

a) Employing Compressed Sparse Row (CSR) format for matrix representation.

b) Implementing incremental disk storage, where computed data batches are immediately written to disk after processing, avoiding prolonged VRAM retention.

3. Tunable Batch Processing: Adjusting the number of rows processed per iteration maximizes GPU utilization while respecting VRAM constraints (e.g., 24GB on NVIDIA RTX 3090).

**Referee Comment 18**

*Line 70, also 243. 'does not surpass 7th order'. Please clarify whether you were using the MCORE code or your own implementation of something similar. Also, is this a fundamental limitation of the mathematical formulation or an issue with a particular implementation? It would be good to clarify what is meant by one-sided interpolation. You could avoid ghost points altogether by doing one-sided reconstruction, but I don't think that is what you mean. Do you mean that with one-sided interpolation there is no coupling between ghost points on the two sides of a panel boundary?*

**Response to Referee:**

The previous statement was indeed ambiguous. What we intended to convey here is that during the design of the ghost cell interpolation for HOPE, we initially attempted to use a one-sided reconstruction stencil similar to MCORE. While stable integration was achieved with the 3rd-, 5th-, and 7th-order schemes, the model became unstable when schemes of 9th-order or higher were used. In other words, for HOPE, overcoming the 7th-order accuracy limitation necessitated the development of a new ghost cell interpolation scheme.

Therefore, we designed a bilateral interpolation algorithm. This algorithm employs an iterative procedure that incorporates information from both adjacent panels of the cubed-sphere grid simultaneously. This enabled stable model integration even with higher-order schemes. Though not detailed in the paper, our testing confirmed stable integration even at 13th-order accuracy.

**Referee Comment 19**

*Line 71. I can guess what you mean by ghost interpolation scheme, but many readers will need more explanation at this point, or at least a forward reference to where it is discussed in more detail.*

**Response to Referee:**

In response to your suggestion, we have amended the relevant section of the manuscript as follows:

*"A high-order finite volume model was developed on cubed sphere, named MCORE (Ullrich et al., 2010; Ullrich and Jablonowski, 2012). High-order reconstruction requires information from cells external to panel boundaries (commonly termed ghost cells). Due to coordinate discontinuities across the six panels of the cubed-sphere grid, MCORE implements an interpolation scheme for ghost cells based on one-side information. This approach employs a two-dimensional reconstruction stencil to interpolate prognostic variables onto Gaussian quadrature points within each cell, followed by integration to obtain cell-averaged values."*

**Referee Comment 20**

*Line 84+. The WENO scheme is (or can be) used whenever the model needs to compute a flux. Is that correct? It was not clear to me.*

**Response to Referee:**

If by "whenever" you mean that the WENO algorithm can be employed at any stage within HOPE's computational process, then your understanding is correct. Specifically, HOPE invokes the WENO scheme only once per Runge-Kutta substep to perform all necessary reconstructions. This single invocation yields the reconstructed state fields at all required cell interfaces. Since each cell interface is shared by two adjacent cells, the reconstructed values are then used as input to the Riemann solver to compute the fluxes.

**Referee Comment 21**

*Line 125. It is not yet clear where 'reconstructing' is used in the algorithm, hence this discussion is hard to follow. It would be good to give a brief overview of the method before getting into details. Also, if the reader does not already know what the 'C-property' is then line 125 does not help them. Either explain or omit. It would be worth adding that, although you use \phi_t in the momentum equation, in (13) you still predict \sqrt{G} \phi for mass conservation.*

**Response to Referee:**

We accept your suggestion and have revised the presentation in the manuscript accordingly. Specifically, the description of the 'C-property' has been omitted. While our approach was indeed inspired by related studies investigating this property, our subsequent analysis indicates that the primary cause of numerical oscillations when reconstructing $\sqrt{G}\phi$ directly is topography-induced

discontinuities compromising the smoothness of the ϕ distribution. These discontinuities perturb high-order reconstruction. Therefore, introducing the concept of the 'C-property' at this point is unnecessary. The revised text now reads as follows:

*"To discretize and solve the equation system, we first perform reconstruction on the prognostic variables to obtain their values at the cell interfaces. These reconstructed values are then used within a Riemann solver to compute the numerical fluxes. During the numerical experiments, we observed that reconstructing $\sqrt{G}\phi$ directly leads to non-physical oscillations. This occurs because topography induces discontinuities in the variable ϕ, while high-order reconstruction fundamentally requires smoothness of the field.*

*To address this, inspired by the approach, we instead reconstruct $\sqrt{G}\phi_t$, where $\phi_t = \phi + \phi_s$ is total geopotential height. The detailed formulation of this reconstruction method is presented in Section 3. Crucially, $\sqrt{G}\phi_t$ is used exclusively during the reconstruction step. The prognostic variable remains $\sqrt{G}\phi$ to ensure exact mass conservation."*

**Referee Comment 22**

*Line 138. It could be worth mentioning that, although LMARS is an approximate Riemann solver, it combines two high-order estimates to obtain the flux, so the result is high order.*

**Response to Referee:**

We accept your suggestion. This clarification indeed helps readers better understand the algorithmic design.

**Referee Comment 23**

*Equation (26). A few words of explanation would be helpful. Here we know the \bar{q}_i, since they are predicted by the time stepping, and we wish to determine the coefficients a.*
*To be clear, do we need a version of the matrix R (31) for every grid cell, or is a single matrix R applicable to all grid cells?*

**Response to Referee:**

We acknowledge that your inquiry rightly identifies insufficient clarification regarding the reconstruction matrix $R$ in the manuscript. Crucially, a fundamental advantage of our cubed-sphere grid dynamical core implementation lies in employing a globally shared reconstruction

matrix $R$. This unification signifies that a single instance of $R$ applies identically to all grid cells, thereby:

1. Significantly reducing memory/VRAM requirements

2. Enabling straightforward utilization of PyTorch's conv2d for accelerated reconstruction

To address how $R$ is generated, we have expanded the discussion in the revised manuscript.

**Referee Comment 24**

*Can you clarify whether the 2D WENO scheme is arbitrary order too, or is the implementation currently limited to 3rd and 5th order?. (The namelist file suggests the latter.)*

**Response to Referee:**

In our implementation, two-dimensional WENO schemes with stencil widths of 3, 5, 7, and 9 are available. Support for higher-order schemes is not currently implemented. This is because computing the smoothness indicator β_i, defined as:

$$\beta_i = \sum_{\zeta=1}^{l} \iint_\Omega \left[ \frac{\partial^\zeta}{\partial \hat{x}^{\zeta_1} \partial \hat{y}^{\zeta_2}} p_j(\hat{x}, \hat{y}) \right]^2 d\hat{x} d\hat{y}$$

for schemes of different orders is inherently nonlinear. To enhance computational efficiency, we utilize the symbolic computation capabilities of Mathematica to derive analytical expressions for $\beta_i$ directly from the reconstruction polynomial $p_j(\hat{x}, \hat{y})$. Consequently, natively supporting arbitrarily high-order WENO in Fortran or PyTorch would necessitate the direct integration of these symbolically computed results into the model code. Achieving this would likely require significant additional engineering development costs, which we deem disproportionate to the benefits gained. First, WENO schemes of orders 3, 5, 7, and 9 are generally sufficient for the requirements of atmospheric simulations in the vast majority of cases. Moreover, the number of required sub-stencils increases substantially with the order of the scheme. For a 9th-order 2D WENO scheme, the number of sub-stencils reaches 25. Even when employing symbolic computation to directly provide the expression for $\beta_i$, the computational burden increases rapidly with the order of the scheme.

**Referee Comment 25**

*Lines 225 to 229. This section is confusing: You define \gamma^+ and \gamma^- then jump to expressing q(x,y) in terms of \omega^+ and \omega^-.*

**Response to Referee:**

We acknowledge that the original presentation exhibited a conceptual discontinuity between defining $\gamma^+/\gamma^-$ and the expression for $q(x, y)$. Our intention was to first present the complete formulation of the WENO scheme before defining its components. As you correctly noted, this approach created an unjustified logical leap.

To address this, we have restructured the section. We first introduce how WENO assigns nonlinear weights $\omega_i$, ( $i = 1,2, ..., s$) to candidate stencils based on smoothness measurements to maintain the Essentially Non-Oscillatory property. We subsequently present the calculation procedure for the weights $\omega_i$. Finally, we provide the expression for the reconstructed value $q(x, y)$.

This sequential presentation establishes clearer logical progression from concept to implementation.

**Referee Comment 26**

*Line 238. '...eight panel boundaries...'. Please check!*

**Response to Referee:**

We sincerely apologize for the incorrect statement regarding the number of cubed-sphere panel boundaries. You are absolutely right – a cube has 12 edges, not eight. This was an oversight that should not have occurred in a rigorous scientific manuscript. We have corrected this error in the revised manuscript. We greatly appreciate your meticulous review, which has improved the precision of our work.

**Referee Comment 27**

*The scheme for ghost cell interpolation neatly exploits the auto-differentiation capability of PyTorch! What do you do near panel corners? Section 3.3.1: could you please clarify, is the iterative scheme used once at setup to obtain the matrix G, and then matrix multiplication is used subsequently at run time? Presumably there can be lots of zeros in G, since cells near the centre of a panel do not affect any ghost cells; thus, could some compact representation of G be used?*

**Response to Referee:**

Your observation is quite astute regarding the necessity for special handling near the corners of the cubed-sphere grid. However, this process does not introduce significant complexity. Similar to the approach used near panel boundaries, we extend the reconstruction stencil. This leverages information from the adjacent panel to populate the ghost cells, thereby ensuring each cell possesses a complete square-shaped reconstruction stencil.

It is important to note that overlapping GQPs occur at the corner positions of the cubed-sphere grid, as illustrated by the magenta points in Figure 2(b). These points lie on the interface shared by adjacent panels (e.g., Panel 1 and Panel 5). Consequently, during ghost value interpolation, two distinct interpolated values are obtained at these overlapping points – one from each adjoining panel. The final interpolated value is computed as the average of these two values. Since the interpolation performed on each individual panel is high-order, the approximation order is preserved when taking this average.

[Figure]

**Figure 3** (a) Adjacent area of panels 1,4 and 5. (b) $5 \times 5$ reconstruction stencil nearby panel corner is represented by shade. The cell contains red dot is the target cell on panel 4; the magenta points are overlapped GQPs shared by panel 1 and panel 5; red solid lines are boundary of panel 4, red dash lines are extension line of panel 4 boundary line. $A$ and $C$ are points on dash line, $B$ is the upper right corner point of panel 4.

$\mathcal{G}$ is a sparse matrix containing many zero entries. To avoid unnecessary memory costs, we adopt the Compressed Sparse Row (CSR) format for storing $\mathcal{G}$. Furthermore,the size of $\mathcal{G}$ is extremely large,making direct application of $torch.autograd.functional.jacobian$ to generate $\mathcal{G}$ computationally infeasible. our implementation for generating ghost cell interpolation matrix achieves significant acceleration and substantially reduces VRAM demand compared to PyTorch's native "$torch.autograd.functional.jacobian$" function. The key optimizations are:

1. Parallel Multi-Row Computation: Utilizing "$torch.vmap$" to encapsulate "$torch.func.vjp$", enabling simultaneous computation of multiple matrix rows.

2. CSR Compression & Incremental Disk Storage:

a) Employing Compressed Sparse Row (CSR) format for matrix representation.

b) Implementing incremental disk storage, where computed data batches are immediately written to disk after processing, avoiding prolonged VRAM retention.

3. Tunable Batch Processing: Adjusting the number of rows processed per iteration maximizes GPU utilization while respecting VRAM constraints (e.g., 24GB on NVIDIA RTX 3090).

**Referee Comment 28**

*Line 312. Comment: the Wicker-Skamarock RK scheme is 3rd order only for linear problems. Which scheme was used to produce figure 8? Was it one of the WENO schemes or the arbitrary order (non-WENO) scheme? In either case, what was the order of accuracy? It is encouraging that there are no numerical oscillations in the vorticity or other fields. Is that true for all the schemes discussed, or only for the WENO schemes?*

**Response to Referee:**

Figure 8 presents simulation results obtained using the TPP5 scheme. We tested the zonal flow case with TPP3, TPP5, TPP7, WENO3, and WENO5 schemes. All tested schemes produced similar results. The use of WENO versus non-WENO reconstruction did not yield a fundamental difference in the outcome. The primary factor significantly impacting the solution quality was the choice of reconstruction variable. Specifically, reconstructing $\sqrt{G}\phi_t$ instead of directly reconstructing the prognostic variable $\sqrt{G}\phi$ proved crucial. This is because the distribution of $\sqrt{G}\phi$ becomes discontinuous due to the presence of topography, whereas $\sqrt{G}\phi_t$ remains

relatively smooth. This critical aspect of the formulation, which addresses this issue, is detailed in Eqs. (13)-(14) of Section 2.

*Points related to equations and mathematical notation*

*Referee Comment 29*

*Equation (21). r is a dummy subscript in the middle expression; it should not appear in the final expression. Similarly for equation (22).*

**Response to Referee:**

We thank the reviewer for identifying this error. The corrections have been implemented in the manuscript.

**Referee Comment 30**

*Line 162-163 does not make sense, since you have not specified any relation between k and n. There are many inconsistencies in notation in this section. n is the number of terms in a polynomial (line 162), then it is the stencil width in the x-direction (23). m is the stencil width in the y-direction (23), then it is equal to n^2 (line 169, 207, 213). k is the width of the stencil (line 163) then a dummy index for coefficients (23). Line 207: the stencil width is now h (but h is not mentioned again). In section 4 the stencil width is s_w.*

**Response to Referee:**

We acknowledge the inconsistencies in notation raised by the reviewer. We have implemented the following corrections throughout the manuscript and verified consistency in relevant sections:

1. **TPP Reconstruction (Section 3):**

(1) The stencil width is now uniformly denoted as $n$.

(2) The total number of cells within the stencil is denoted as $N$.

**2. WENO Algorithm (Section 3):**

(1) $h$ specifically denotes the width of the high-order stencil.

(2) $l$ specifically denotes the width of the low-order stencil.

(3) This distinction was intentionally introduced to emphasize the difference in scale between these stencils, even when numerically identical. Although h and l may not be explicitly

referenced later, the notation serves to prevent conceptual conflation of stencils with different intended orders of accuracy.

**3. Section 4 ($s_w$):**

(1)  $s_w$ represents the stencil width within the context of the PyTorch code implementation.

(2)  This notation was adopted directly from the HOPE-PyTorch codebase to enhance clarity and facilitate code comprehension for readers.

(3)  We acknowledge the objectively valid inconsistency with the notation in Section 3. Therefore, we have added an explicit statement in the manuscript equating $s_w$ with n (the stencil width defined in Section 3).

**Referee Comment 31**

*Inconsistent fonts are used for the matrix \gamma (compare (32) and (35)).*

**Response to Referee:**

We appreciate this observation and have standardized the notation for vector $\boldsymbol{\gamma}$ throughout the manuscript.

**Referee Comment 32**

*Presumably (36) refers to individual elements of the matrix \gamma, not the entire matrix? (37) and (38) don't seem to be correct. (38) implies that \sum_i \omega_i^+ = 1 and \sum_i \omega_i^- = 1. However, in order for (37) to be a proper weighted average of the p_i's we would need \sum_i (\omega_i^+ - \omega_i^-) = 1. \omega_i is mentioned in the text, but only \omega_i^+ and \omega_i^- are defined by equations. Should we assume \omega_i = \omega_i^+ - \omega_i^- ? Please check.*

**Response to Referee:**

This formulation indeed contains an omission, as noted by the referee.

The reconstruction value on point $(x, y)$ is expressed by:

$$q(x, y) = \sum_{i=1}^{s} (\sigma^+ \omega_i^+ - \sigma^- \omega_i^-) p_i(x, y)$$

where

$$\sigma^{\pm} = \sum_{i=1}^{s} \widetilde{\gamma}_i^{\pm}$$

and

$$\widetilde{\boldsymbol{\gamma}}^+ = \frac{\theta|\boldsymbol{\gamma}| + \boldsymbol{\gamma}}{2}, \qquad \widetilde{\boldsymbol{\gamma}}^- = \boldsymbol{\gamma}^+ - \boldsymbol{\gamma}$$

$\widetilde{\gamma}_i^{\pm}$ is the i-th element of $\widetilde{\boldsymbol{\gamma}}^{\pm}$, $\gamma_i^{\pm}$ is the i-th element of $\boldsymbol{\gamma}^{\pm}$

We have accordingly corrected the relevant formulas and refined their formulation in the manuscript.

**Referee Comment 33**

*Equation (53) seems to be dimensionally inconsistent. Should sign(m) not be abs(m), which would pick out the upwind value of q? See also line 346. Actually, taking careful note of parentheses, the (fortran) source code seems to be correct, but is inconsistent with equation (53).*

**Response to Referee:**

Thanks a lot for finding out the mistake!

Eq.(53) should be:

$$\boldsymbol{F} = \frac{1}{2} m[(\boldsymbol{q}_L + \boldsymbol{q}_R) - sign(m)(\boldsymbol{q}_R - \boldsymbol{q}_L)] + \boldsymbol{P}$$

The wind speed $m$ should be multiplied outside the brackets. Of course, replacing $sign(m)$ by $abs(m)$ is also correct.

**Referee Comment 34**

*The notation G is used for the metric (section 2) and also for the matrix to compute ghost cell values (section 3.3.1).*

**Response to Referee:**

This notation conflict has been identified. To resolve the ambiguity, we now denote the ghost cell interpolation matrix with a calligraphic symbol $\mathcal{G}$ throughout the manuscript.

**Referee Comment 35**

*Line 262: g should be bold font.*

**Response to Referee:**

The notation has been corrected to use boldface for $\boldsymbol{g}$ in the manuscript.

**Referee Comment 36**

*Lines 324 and 327: can I just check that there should be no comma between n_v and n_p, i.e., the first dimension is of size n_v \times n_p? The code (if I understand it correctly) suggests that these arrays are 5-dimensional. Also, comparing lines 327 and 334, n_{poc} seems to be the size of the second (or perhaps third) dimension, not the first.*

**Response to Referee:**

We acknowledge that the issues you identified accurately reflect ambiguities in the tensor dimension descriptions within the manuscript. Consequently, we have restructured the relevant section detailing the Gaussian quadrature scheme.

Briefly stated, the rationale for employing einsum rather than matmul arises because the dimension containing quadrature points is not the last dimension following reconstruction via conv2d, thereby precluding direct use of matmul for Gaussian quadrature.

Your observation is particularly astute: during the reconstruction process using the conv2d function, the shape of $\boldsymbol{q}$ is transformed from $(n_v, n_p, 1, n_c, n_c)$ to $(n_v n_p, 1, n_c, n_c)$. This reshaping occurs because conv2d requires the first dimension to represent batch size and the second channel size, necessitating the collapse of $n_v$ and $n_p$ into a unified batch dimension.

*Points related to phrasing, typos, etc*

**Referee Comment 37**

*Line 17, line 53. 'intensive' panel boundary treatment. What is meant by intensive? Perhaps a different word would be better?*

**Response to Referee:**

Thank you for the suggestion. During the initial design phase, we intended the term "intensive" to convey that the new ghost cell interpolation scheme incorporates information from both sides of the panel boundary, while requiring a narrower halo region compared to one-sided reconstruction. We agree that "two-way coupling" provides a more accurate description, as it directly conveys the core characteristic of the new interpolation algorithm. Consequently, we have revised all relevant instances in the manuscript to use the new descriptor: "two-way coupled panel boundary."

**Referee Comment 38**

*Line 34. 'Unlike...' is not a complete sentence. Perhaps the preceding full stop should be a comma?*

**Response to Referee:**

You are correct; this was a punctuation error. We have replaced the period with a comma in the revised manuscript.

**Referee Comment 39**

*Line 78. Does 'its' refer to the new ghost interpolation scheme?*

**Response to Referee:**

We clarify that "its" refers to the effectiveness of the Taylor series expansion itself, not the ghost interpolation scheme. Our statement highlights fundamental mathematical constraints for Taylor series approximations:

The accuracy of approximating a function via a Taylor series requires two essential conditions:

(1) The existence of higher-order derivatives of the function at the expansion point,

(2) The convergence of the series within the relevant domain.

This context was intended to motivate the limitations of Taylor-based reconstructions in discontinuous regions, which necessitate the non-oscillatory schemes discussed subsequently.

**Referee Comment 40**

*Line 92. If I understand correctly, GPU optimization and automatic differentiabilty are two different things; PyTorch happens to provide them both. The sentence as written implies that automatic differentiation is needed for GPU implementation, which I don't think is correct.*

**Response to Referee:**

You are correct. Our original wording inaccurately implied a dependency between GPU implementation and automatic differentiation. The revised text now clarifies: "Section 4 focuses on HOPE's high-performance implementation leveraging PyTorch's built-in operators for GPU acceleration. The adoption of PyTorch simultaneously enables automatic differentiation

capabilities through its computational graph construction." This modification explicitly distinguishes the two independent features provided by PyTorch.

*Line 133. Can you clarify: Gaussian quadrature along the interface (rather than, say, over some upwind region).*

**Response to Referee:**

The purpose of Gaussian quadrature is to compute the line integral of fluxes along cell edges. As shown in Eqs. (17)-(20), the finite-volume discretization requires that the tendency of cell-averaged quantities depends on the net flux through cell boundaries. Since obtaining an analytic expression for the flux distribution along the entire edge is impractical (Riemann solvers typically operate on pointwise reconstructed states rather than continuous flux functions), Gaussian quadrature provides high-order integration: for smooth flows, $m$ quadrature points achieve $(2m - 1)$th-order accuracy, enabling precise numerical integration of interfacial fluxes.

Regarding your query about upwind regions: While reconstruction stencils are symmetric within each cell, asymmetry arises at shared interfaces. Specifically: Left-biased reconstruction incorporates more information from the left cell; Right-biased reconstruction incorporates more from the right cell. The Riemann solver resolves this by assigning greater weight to information from the upstream side based on characteristic wave propagation directions.

Your reference to integrating over an upwind region aligns conceptually with Flux-Form Semi-Lagrangian (FFSL) methods. While currently not implemented in HOPE, we acknowledge their efficiency and plan to explore FFSL for tracer advection in future developments.

**Referee Comment 42**

*Line 162. The term 'order' is already overloaded. It is not necessary to talk about a k'th-order square stencil. It is enough to say k \times k stencil. See also line 206.*

*Line 163: n^2 is the number of cells in the stencil ('cell number in the stencil' is ambiguous). Similarly, it is the number of terms in the TPP.*

**Response to Referee:**

We have restructured Section 3.1 and 3.2 in response to this comment. The revisions clarify the symbolic expressions and standardize the terminology used to describe stencil sizes throughout the section.

*Line 210. The phrase 'determine the unique weights' suggests that (32) can be solved and has a unique solution. As soon becomes clear, (32) is overdetermined and has no exact solution, and only a least squares approximate solution can be found.*

**Response to Referee:**

Your analysis is entirely valid. We would like to share an interesting finding regarding Eq. (32) (In new version manuscript Eq.(34)). While the system appears overdetermined at first glance, solving it via the least squares method yields a unique solution. Furthermore, substituting this solution back into Eq. (32) reveals that it satisfies the linear system exactly.

This discovery was a significant conclusion during the development of HOPE. Initially, when designing the 2D WENO scheme (around year 2021), we were skeptical about the existence of optimal weights precisely due to the overdetermined nature of the linear system, as you point out. Proceeding experimentally, we investigated the case for WENO5 (WENO with stencil width 5). Applying the least squares method to solve the system yielded a set of weights that appeared remarkably concise, physically plausible, and, intuitively, almost too specific. To our surprise, substituting these weights back into the equations demonstrated that they satisfied Eq. (32) exactly.

Subsequent review of WENO literature revealed that this phenomenon is not unprecedented. Indeed, earlier research on 1D WENO and WENO on Triangular meshes (Hu and Shu, 1999) also leveraged this very property.

*Line 227. 'stencil i is smooth ... stencil i is discontinuous...'. Don't you mean the data sampled or reconstructed on stencil i is smooth or discontinuous?*

**Response to Referee:**

We have change the description in new manuscript:

"*The WENO scheme adaptively assigns nonlinear weights $\omega_i, (i = 1,2,...,s)$ to each candidate stencil to suppress unphysical oscillations during high-order reconstruction. Essentially, it gives greater weight to stencils identified as smooth over the local cell, while suppressing the influence of those containing discontinuities by assigning them smaller weights.*"

**Referee Comment 45**

*Line 301. 'location, since'. Full stop and a new sentence would be better.*

**Response to Referee:**

We appreciate your diligent review of the manuscript. This specific correction has been implemented in the revised version of the paper as suggested.

**Referee Comment 46**

*Equation (47). Since Einstein summation is mentioned in various places, perhaps note that there is no summation over i in (47).*

**Response to Referee:**

Your suggestion is crucial for avoiding ambiguity regarding the meaning of the superscripts and aids readers in correctly interpreting Equation (47) (Equation (51) in the revised manuscript). We have added an explicit clarification to this effect in the new version of the paper.

**Referee Comment 47**

*Line 310. I cannot find any other mention of H.*

**Response to Referee:**

It should be $\boldsymbol{G}$ (the flux in y direction), we have revised the presentation in the manuscript accordingly.

**Referee Comment 48**

*Line 318-321. 'Both of these operations are highly optimized for execution on GPUs...' Do you mean highly optimized in the PyTorch implementation? The next sentence seems to be confusing two distinct ideas: (i) PyTorch has built-in commands for convolutions and matrix-vector*

*multiplication, streamlining implementation (without explicit loop commands); (ii) PyTorch offers*

*automatic differentiation, enabling efficient gradient computation.*

**Response to Referee:**

To enhance clarity, we have revised the relevant passage in response to this point.

"*The spatial operator and temporal integration of HOPE can be easily implemented using*

*PyTorch. Specifically, the spatial reconstruction given by Eq.错误!未找到引用源。 is*

*implemented as a convolution operation, while the Gaussian quadrature can be efficiently*

*expressed as a matrix-vector multiplication. Leveraging PyTorch's highly optimized built-in*

*functions for both convolution and quadrature operations ensures superior performance on GPUs.*

*Furthermore, PyTorch's role as a versatile AI development platform provides automatic*

*differentiation capabilities across the entire computation graph. This streamlines implementation*

*and enables efficient gradient computation for all components.*"

**Referee Comment 49**

*Line 323. To be clear, n_v prognostic variables per cell.*

**Response to Referee:**

Good advice, this specific correction has been implemented in the revised version of the paper as suggested.

**Referee Comment 50**

*Line 359: widely?*

**Response to Referee:**

We appreciate your diligent review of the manuscript. This specific correction has been implemented in the revised version of the paper as suggested.

**Referee Comment 51**

*Line 363. You haven't said what \alpha is, other than a number that is set to zero.*

**Response to Referee:**

In the steady-state flow test, $\alpha$ represents the orientation angle of the flow field relative to due

east. While Williamson et al. (1992) explore various values of $\alpha$ in their benchmark specifications,

our sensitivity analysis confirms that the choice of $\alpha$ does not affect the measured convergence rate. Consequently, we present accuracy results exclusively for the $\alpha = 0$ configuration in our convergence study.

**Referee Comment 52**

*Line 387. The phrase 'we set' makes it seem like you have made your own choice for \lambda_c and \theta_c. But aren't those values the standard ones for this test case?*

**Response to Referee:**

Thank you for this observation. Our phrasing here was inaccurate and potentially misleading. The values $\lambda_c = \frac{3\pi}{2}, \theta_c = \frac{\pi}{6}$ are indeed the standard values specified for this test case in Williamson et al. (1992). We have corrected the corresponding statement in the manuscript accordingly.

**Referee Comment 53**

*Line 401. zonal advection -> zonal propagation. (The wave structure is not simply advected in the zonal direction; it propagates through the Rossby wave propagation mechanism.)*

**Response to Referee:**

We acknowledge the imprecise terminology in the original text. Your characterization more accurately describes the Rossby wave propagation mechanism, and we have implemented this correction throughout the manuscript.

**Referee Comment 54**

*Line 404. Please check the units for c.*

**Response to Referee:**

We thank the reviewer for identifying this error, "day" should be "days".

**Referee Comment 55**

*Line 481. 'handling of anomalous anisotropic characateristics'. I think the problem is that the 1D scheme lacks isotropy, rather than the data.*

**Response to Referee:**

Your observation regarding this terminological distinction is crucial. We acknowledge that the original phrasing was inaccurate and ambiguous; this section has been rectified in the revised text as:

"*Therefore, when simulating fluid fields characterized by isotropic features, the 1D scheme lacks the capability to accurately represent diagonal directional features. Conversely, the 2D scheme correctly captures the inherent isotropic characteristics.*"

**Referee Comment 56**

*References: Kochkov et al.; the year should be 2024.*

**Response to Referee:**

We appreciate your diligent review of the manuscript. This specific correction has been implemented in the revised version of the paper as suggested.

---

## Author Comment (AC3)

**Referee Comment**

- *Also, it is good to be aware of what the time of collapse is really telling you about the model formulation, and to look at other informative aspects of the solution. For example, you mention apparent 'dissipation' of the solution. You could measure that dissipation quantitiatively by diagnosing a conserved quantity like energy or potential enstrophy, for example, and look at how their conservation depends on resolution and order of accuracy.*

**Response to Referee:**

Your suggestion is very well-founded. Therefore, we have incorporated a new figure into the manuscript depicting the time series of total energy, total potential enstrophy and total zonal angular momentum. The following addition has been made to the manuscript:

We measure the conservation errors by defining the normalized error $\epsilon_r$ of the variable $\eta$ as

$\epsilon_r = \frac{I_g(\eta^n) - I_g(\eta^0)}{I_g(\eta^0)}$, where $\eta^0$ and $\eta^n$ stand for $\eta$ value at initial time and time slot $n$, respectively.

The global integral is defined as:

$$I_g(\eta^n) = \sum_{p=1}^{n_p} \sum_{j=1}^{n_y} \sum_{i=1}^{n_x} \sqrt{G}_{i,j,p} \bar{\eta}_{i,j,p}$$

where $\bar{\eta}_{i,j,p}$ represents the average value of $\eta$ in cell $(i,j,p)$

A 100-day simulation of the Rossby-Haurwitz wave was conducted using a C90 grid (1° resolution). The total energy simulated with the TPP3, TPP5, TPP7, and TPP9 schemes underwent dissipation to varying degrees. By day 100, the normalized total energy errors reached $-1.49 \times 10^{-3}, -1.33 \times 10^{-5}, -1.71 \times 10^{-6}, -4.20 \times 10^{-7}$, respectively, indicating significantly stronger dissipation for the TPP3 scheme compared to the other higher-order schemes Figure 1 (a)。 Figure 1 (b) presents a scaled view of the energy evolution for TPP5, TPP7, and TPP9, clearly demonstrating that increasing the reconstruction order progressively reduces energy dissipation. Furthermore, following the RH wave collapse, a significant drop in total energy was observed for the TPP5 scheme (after approximately 90 days) and the TPP7 scheme (after approximately 95 days).

[Figure]

**Figure 1** Time series of normalized conservation errors for the Rossby-Haurwitz wave simulation on the C90 grid over days 0 to 100, with LMARS scheme as Riemann solver. (a) Normalized total energy error for TPP3, TPP5, TPP7 and TPP9. (b) The total energy normalized error for TPP5, TPP7 and TPP9. (c) Normalized potential enstrophy error for TPP3, TPP5, TPP7 and TPP9. (d) Normalized total zonal angular momentum error for TPP3, TPP5, TPP7 and TPP9.

Analysis of the normalized total potential enstrophy error (Figure 1 (c)) and the normalized zonal angular momentum error (Figure 1 (d)) over time yields conclusions consistent with those for total energy. Specifically, the TPP3 scheme exhibited substantially higher dissipation than the higher-order schemes, confirming that employing higher-order reconstruction schemes effectively minimizes dissipation. Notably, significant dissipation surges occurred in these quantities following the RH wave collapse.

---

## Author Comment (AC7)

We sincerely appreciate your questions and suggestions. Based on the issues you identified, we have implemented significant revisions to the manuscript. We commend your scientific rigor throughout this process.

**Referee Comment 1**

*Line 50: The model uses a finite-volume scheme. Please explain why it is characterized as a local-stencil-based model.*

**Response to Referee:**

Thank you for your suggestion. The term "local stencil-based model" refers to the computational characteristic of our finite-volume scheme, where updating a cell's state during explicit time marching requires only information from a local stencil surrounding that cell. This design eliminates the need for global communication, making each time step highly parallelizable. Accordingly, we have revised the sentence to:

"A finite-volume scheme requiring only information from a local stencil surrounding each cell to perform state updates, enabling massively parallel scalability."

**Referee Comment 2**

*Line 52: It should be explicitly stated that the attractive property discussed here arises from the WENO scheme.*

**Response to Referee:**

According to your advice, the new description is written as:

"A WENO (Weighted Essentially Non-Oscillatory) based, adaptive polynomial order reduction for essential non-oscillation."

**Referee Comment 3**

*Figure 4: Panel (b) shows the spatial stencil for a quadratic polynomial, and panel (c) for a quartic polynomial. Please correct the caption.*

**Response to Referee:**

The new version of Figure 4 (Figure 1 in this reply file) is shown as

[Figure]

**Figure 1** Reconstruction coordinate and polynomial terms on stencils. (a): Local reconstruction coordinate (the red points denote cell centers) (b): 2nd degree polynomial stencil; (c): TPP3 stencil; (d) TPP5 stencil

HOPE employs an equiangular cubed-sphere grid, where each panel undergoes uniform angular discretization into $n_c \times n_c$ cells. In the computational space (equiangular coordinates), each cell spans an angular interval of $\frac{\pi}{2nc}$, therefore

$$\Delta x = \Delta y = \frac{\pi}{2nc}$$

This uniformity ensures that all cells are geometrically identical in the computational space, thereby avoiding the need for cell-specific treatment during reconstruction studies. In the following part of this section, we set a new computational space for reconstruction process. The coordinate system $(\hat{x}, \hat{y})$ is established such that within each reconstruction stencil, the origin $(0,0)$ is located at the stencil center, the central cell spans$[-0.5, 0.5]$ in both $\hat{x}$ and $\hat{y}$ directions, as shown in Figure 1 (a). All of the cells have the same size in $\hat{x}, \hat{y}$ directions:

$$\Delta \hat{x} = \Delta \hat{y} = 1$$

On the cubed-sphere grid, a fixed reconstruction scheme yields consistent stencils across all cells. This structural homogeneity renders the reconstruction operation computationally equivalent

to two-dimensional convolution, thereby enabling efficient GPU acceleration through PyTorch's built-in conv2d function.

To construct genuinely 2D reconstructions, the functional form of the reconstruction basis must be selected. A bivariate polynomial of degree $d$ contains $\frac{(d+1)(d+2)}{2}$ terms. As illustrated in Figure 1 (b), the 6 terms of a bivariate quadratic polynomial ($d = 2$) are insufficient to cover a square stencil. To address this, we adopt Tensor Product Polynomials (TPP) as basis functions. We denote a TPP function containing $n \times n$ terms as TPPn. Determining the coefficients of TPPn requires information from a $n \times n$ block of cells. When using a TPP reconstruction stencil of size $n \times n$, HOPE achieves fifth-order accuracy when simulating smooth flow fields. We therefore designate a TPP reconstruction stencil of size $n \times n$ as an $n$-th order TPP stencil, the 3$^{\text{rd}}$ and 5$^{\text{th}}$ order TPP stencils are shown in Figure 1 (c)(d).

**Referee Comment 4**

*Figure 7: The ghost cells are interpolated using a two-dimensional procedure, which involves solving a system of equations iteratively. I recommend the authors consider employing a one-dimensional interpolation scheme instead, as the quadrature points in ghost cells are arranged along lines connecting the corresponding points in neighboring cells. One-dimensional interpolation can simplify the interpolation and improve efficiency.*

**Response to Referee:**

Thank you for your suggestion. While a one-dimensional interpolation scheme would indeed be efficient for models employing a dimension-by-dimension reconstruction approach, our testing indicates that a 1D ghost cell interpolation scheme cannot achieve accuracy beyond second order—consistent with findings from Ullrich et al. (2010). This limitation arises because HOPE integrates a two-dimensional reconstruction scheme with cell-boundary flux calculations. Using 1D reconstruction for ghost cell interpolation would cause a loss of two-dimensional information.

To illustrate, consider reconstruction along the x-direction: a pure 1D scheme computes ghost point values not as true pointwise quantities, but as integral averages along the y-direction within the cell. Recovering the actual point values would require an additional deconvolution operation

(Ullrich et al., 2010). Crucially, this process necessitates a wider interpolation stencil than the original scheme.

As you noted, our proposed ghost cell interpolation method appears to require solving linear systems for a closed-form expression. We acknowledge in the Appendix that direct inversion of such large-scale systems is impractical. HOPE's key innovation (Section 3.3.1) circumvents this by leveraging PyTorch's auto-differentiation:

1. The ghost cell interpolation matrix $\mathcal{G}$ is computed *row-wise* via automatic differentiation.

2. $\mathcal{G}$ is stored efficiently in Compressed Sparse Row (CSR) format.

3. Ghost cell interpolation then reduces to a single matrix-vector multiplication.

This approach dramatically reduces computational costs while preserving high-order accuracy.

**Referee Comment 5**

*Subsection 3.5: As the model is based on a WENO scheme, I recommend using a TVD Runge-Kutta time integration.*

**Response to Referee:**

Thank you for the valuable suggestion. The pairing of WENO with a TVD Runge-Kutta time integration scheme is indeed appropriate. HOPE is compatible with various explicit Runge-Kutta schemes, and our experiments have comprehensively evaluated multiple such methods. For the test cases presented in this paper, results demonstrate no discernible differences between using TVD Runge-Kutta and the WRF Runge-Kutta formulation.

**Referee Comment 6**

*I suggest including results for solid rotation of a cosine bell along different directions. Please also provide time histories of normalized errors.*

**Response to Referee:**

The Solid Body Rotation Cosine Bell (Case 1 from Williamson (1992)) is widely used to assess noise generated by panel boundaries, as noted by Chen and Xiao (2008), Ullrich et al. (2010).

Figure 2 presents the norm errors for a 12-day simulation at $\alpha = 0$; results for $\alpha = \pi/2$ are identical. The temporal evolution of $L_1$ and $L_2$ norm errors does not exhibit a pronounced signature attributable to panel boundaries. In contrast, the $L_\infty$ norm error evolution shows significant sensitivity to panel boundaries, varying considerably with grid resolution and reconstruction order. When using low resolution and low reconstruction order (TPP3 with C30 grid), oscillations induced by panel boundaries are relatively weak. However, as the model resolution or reconstruction order increases, the influence of panel boundaries on the $L_\infty$ norm error manifests as a distinct four-peak pattern, corresponding to the four longitudinally aligned panel boundaries of the cubed-sphere grid.

[Figure]

**Figure 2** The variation of norm errors during simulation days for the cosine bell advection test case, with direction parameter $\alpha = 0$. The rows represent reconstruction schemes TPP3, TPP5 and TPP7, the columns stand for grid C30, C45, C90 and C180.

Figure 3 shows the 12-day simulation norm errors for $\alpha = \pi/4$. In this test configuration, the cosine bell initially moves alone the interface between Panel 1 and Panel 5, and subsequently moves along the interface between Panel 3 and Panel 6. The temporal evolution of $L_1$ and $L_2$ norm errors display two gentle peaks, corresponding to the errors generated as the cosine bell crosses

these panel interfaces. Similar to Figure 2, the $L_\infty$ norm error progressively exceeds the $L_1$ and $L_2$ norm errors as grid resolution and reconstruction order increase.

[Figure]

**Figure 3** The variation of norm errors during simulation days for the cosine bell advection test case, with direction parameter $\alpha = \pi/4$. The rows represent reconstruction schemes TPP3, TPP5 and TPP7, the columns stand for grid C30, C45, C90 and C180.

Because the Cosine Bell field lacks infinite continuity, the convergence rate of the norm errors cannot exceed second order in our tests, regardless of the reconstruction order employed. This observation aligns with the key point emphasized in our paper: high-order numerical methods achieve their design accuracy only when the flow field is sufficiently smooth. Discontinuities in the flow field violate the fundamental premise of polynomial reconstruction (as discontinuities impair the continuity of higher derivatives, leading to non-convergence of the Taylor series). This inherent sensitivity to smoothness is precisely the factor causing norm errors to be influenced by cubed-sphere panel boundaries. When using low-order reconstruction schemes at low resolutions, the Tensor Product Polynomial (TPP) reconstruction employs lower-degree polynomials and is consequently less sensitive to the smoothness of the flow field. Conversely, high-order TPP reconstruction requires the flow field to possess higher-order continuity to maintain accuracy; it is thus more sensitive to discontinuities. Insufficiently smooth flow fields can introduce numerical

oscillations with high-order schemes. Therefore, while TPP5 and TPP7 yield lower $L_\infty$ norm error magnitudes than TPP3, they exhibit more pronounced oscillations caused by the cubed-sphere panel boundaries.

**Referee Comment 7**

*Williamson test case 2: It would also be helpful to present results obtained using the corresponding linear scheme (i.e., by applying optimal weights in WENO schemes directly). Displaying the absolute error distributions will be helpful to evaluate the grid imprinting.*

**Response to Referee:**

Thank you for your valuable suggestion. We have visualized the simulation errors and obtained meaningful insights.

As shown in Figure 4, errors near the panel boundaries of the cubed-sphere grid are significantly higher than those in the central regions, confirming the presence of grid imprinting. Furthermore, we implemented the AUSM$^+$-up Riemann solver (consistent with the scheme described in Ullrich et al. (2010)) as an alternative to LMARS. While computationally more complex, AUSM$^+$-up substantially reduces simulation errors. Comparative analysis of Figure 4 (a) and (b) demonstrates that the maximum absolute error decreases from $8.792\times10^{-5}$ (LMARS) to $2.4129\times10^{-5}$ (AUSM$^+$-up), while convergence rates remain unchanged.

Performance benchmarks using HOPE's Fortran implementation on a C90 grid show that simulating 12 days with a 200-second integration time step requires 49.4 seconds for LMARS versus 57.34 seconds for AUSM$^+$-up. This demonstrates that Riemann solver selection critically impacts simulation outcomes, consistent with the discussions in Ullrich et al. (2010).

[Figure]

**Figure 4** Numerical errors (simulation result minus exact solution) of geopotential height for steady state flow (Williamson test case 2) with Riemann solvers (a) LMARS and (b) AUSM$^+$-up. The

reconstruction scheme is TPP5.

**Referee Comment 8**

*Williamson test cases 5 and 6: I recommend reporting the time histories of normalized errors of total energy and potential enstrophy.*

**Response to Referee:**

We measure the conservation errors by defining the normalized error $\epsilon_r$ of the variable $\eta$ as $\epsilon_r = \frac{I_g(\eta^n) - I_g(\eta^0)}{I_g(\eta^0)}$, where $\eta^0$ and $\eta^n$ stand for $\eta$ value at initial time and time slot $n$, respectively.

The global integral is defined as:

$$I_g(\eta^n) = \sum_{p=1}^{n_p} \sum_{j=1}^{n_y} \sum_{i=1}^{n_x} \sqrt{G}_{i,j,p} \bar{\eta}_{i,j,p}$$

where $\bar{\eta}_{i,j,p}$ represents the average value of $\eta$ in cell $(i, j, p)$

In the 15-day simulation of zonal flow over an isolated mountain the total energy exhibited a gradual increase over the integration time, while both the total potential enstrophy and the total zonal angular momentum showed gradual dissipation as the simulation progressed. The AUSM⁺-up scheme demonstrated stronger energy dissipation compared to the LMARS scheme, as illustrated in Figure 5。

[Figure]

**Figure 5** Time series of normalized conservation errors for the zonal flow over isolated mountain simulation on the C90 grid over days 0 to 100. (a) Normalized total energy error. (b) Normalized total potential enstrophy error. (c) Normalized total zonal angular momentum error.

A 100-day simulation of the Rossby-Haurwitz wave was conducted using a C90 grid (1° resolution). The total energy simulated with the TPP3, TPP5, TPP7, and TPP9 schemes underwent dissipation to varying degrees. By day 100, the normalized total energy errors reached $-1.49 \times 10^{-3}, -1.33 \times 10^{-5}, -1.71 \times 10^{-6}, -4.20 \times 10^{-7}$, respectively, indicating significantly stronger dissipation for the TPP3 scheme compared to the other higher-order schemes Figure 6 (a)。Figure 6 (b) presents a scaled view of the energy evolution for TPP5, TPP7, and TPP9, clearly demonstrating that increasing the reconstruction order progressively reduces energy dissipation. Furthermore, following the RH wave collapse, a significant drop in total energy was observed for the TPP5 scheme (after approximately 90 days) and the TPP7 scheme (after approximately 95 days).

[Figure]

**Figure 6** Time series of normalized conservation errors for the Rossby-Haurwitz wave simulation on the C90 grid over days 0 to 100, with LMARS scheme as Riemann solver. (a) Normalized total energy error for TPP3, TPP5, TPP7 and TPP9. (b) The total energy normalized error for TPP5, TPP7 and TPP9. (c) Normalized potential enstrophy error for TPP3, TPP5, TPP7 and TPP9. (d) Normalized total zonal angular momentum error for TPP3, TPP5, TPP7 and TPP9.

Analysis of the normalized total potential enstrophy error (Figure 6 (c)) and the normalized zonal angular momentum error (Figure 6 (d)) over time yields conclusions consistent with those for total energy. Specifically, the TPP3 scheme exhibited substantially higher dissipation than the higher-order schemes, confirming that employing higher-order reconstruction schemes effectively minimizes dissipation. Notably, significant dissipation surges occurred in these quantities following the RH wave collapse.

**Referee Comment 9**

*Genuine 2D scheme: The manuscript emphasizes the benefits of using a genuine two-dimensional discretization. The benefits should be demonstrated through Williamson's standard test suite, rather than a dam-break problem, which is not representative of global atmospheric dynamics. Additionally, please quantify the computational cost differences between the dimension-by-dimension and genuinely 2D schemes.*

**Response to Referee:**

The motivation behind designing the dam break test case was specifically to highlight the differences between the genuine two-dimensional scheme and dimension-by-dimension approaches. These differences proved to be understated in Williamson's Case 5 (zonal flow over an isolated mountain) and Case 6 (Rossby-Haurwitz wave).

We recognize that Multi-Moment Finite Volume Method (MCV) algorithms can achieve high accuracy even with dimension-by-dimension discretization. This is because MCV inherently computes the tendency of point values (PV) through its spatial discretization.

Conversely, in the HOPE model, the prognostic variable is the Volume Integral Average (VIA), not PV. When a dimension-by-dimension reconstruction scheme is applied in this context (specifically for Case 2, steady-state geostrophic flow), the accuracy cannot surpass second order. This limitation parallels the situation mentioned in Fig. 2 of Ullrich et al. (2010). The underlying reason, consistent with our response to Referee Comment 4, is that performing a reconstruction in the x-direction targeting the VIA does not directly yield PV. Instead, it produces line integral averages along the y-direction. Consequently, achieving high accuracy necessitates additional convolution or deconvolution operations.

Furthermore, Shi et al. (2002) report in their Table 3.1 that for the simulation of the 2D Vortex Evolution problem, the convergence rates of genuine finite-volume methods and dimension-by-dimension methods are similar. However, the $L_\infty$ error of the genuine method is significantly lower.

In our own performance tests using WENO with a stencil width of 5 on a C90 grid, simulating 1 day with a 200s time step via RK3 TVD, the dimension-by-dimension scheme took 16.4 seconds, while the genuine 2D scheme required 80.9 seconds (Fortran code, Dual E5-2699V4 processors). It is crucial to note that the current research primarily demonstrates the feasibility of the scheme; the model implementation has not undergone high-performance optimization. Therefore, these timing results should be considered preliminary, and significant improvements in computational efficiency are anticipated during future development.

**References**

Chen, C. and Xiao, F.: Shallow Water Model on Cubed-Sphere by Multi-Moment Finite Volume Method, Journal of Computational Physics, 227, 5019-5044, 10.1016/j.jcp.2008.01.033, 2008.

Shi, J., Hu, C., and Shu, C.-W.: A Technique of Treating Negative Weights in WENO Schemes, Journal of Computational Physics, 175, 108-127, 10.1006/jcph.2001.6892, 2002.

Ullrich, P. A., Jablonowski, C., and van Leer, B.: High-Order Finite-Volume Methods for the Shallow-Water Equations on the Sphere, Journal of Computational Physics, 229, 6104-6134, 10.1016/j.jcp.2010.04.044, 2010.

---

## Author Response (AR3)

We note that Referee #2 did not mention any further advice. Once again, we are deeply grateful for their valuable insights and scientific rigor throughout the review process. Their suggestions have undoubtedly led to a more complete and robust manuscript.